

# Retrieving 3D distributions of atmospheric particles using Atmospheric Tomography with 3D Radiative Transfer – Part 2: local optimization

Jesse Loveridge[1], Aviad Levis[2], Larry Di Girolamo[1], Vadim Holodovsky[3], Linda Forster[4], Anthony B. Davis[4], Yoav Y. Schechner[3]

[1]Department of Atmospheric Sciences, University of Illinois at Urbana-Champaign, Urbana, 61801, USA
[2]Computer and Mathematical Sciences Department, California Institute of Technology, Pasadena, 91125, USA
[3]Viterbi Faculty of Electrical and Computer Engineering, Technion – Israel Institute of Technology, Haifa 3200003, Israel
[4]Jet Propulsion Laboratory, California Institute of Technology, Pasadena, 91109, USA

*Correspondence to*: Jesse Loveridge (jesserl2@illinois.edu)

**Abstract.** Our global understanding of clouds and aerosols relies on the remote sensing of their optical, microphysical, and macrophysical properties using, in part, scattered solar radiation. Current retrievals assume clouds and aerosols form plane-parallel, homogeneous layers and utilize 1D radiative transfer (RT) models. These assumptions limit the detail that can be retrieved about the 3D variability of cloud and aerosol fields and induce biases in the retrieved properties for highly heterogeneous structures such as cumulus clouds and smoke plumes. In Part 1 of this two-part study, we validated a tomographic method that utilizes multi-angle passive imagery to retrieve 3D distributions of species using 3D RT to overcome these issues. That validation characterized the uncertainty in the approximate Jacobian used in the tomographic retrieval over a wide range of atmospheric and surface conditions for several horizontal boundary conditions. Here in Part 2, we test the algorithm's effectiveness on synthetic data to test whether retrieval accuracy is limited by the use of the approximate Jacobian. We retrieve 3D distributions of volume extinction coefficient ($\sigma_{3D}$) at 40 m resolution from synthetic multi-angle, mono-spectral imagery at 35 m resolution derived from stochastically-generated 'cumuliform' clouds in (1 km)³ domains. The retrievals are idealized in that we neglect forward modelling and instrumental errors with the exception of radiometric noise; thus reported retrieval errors are lower bounds. $\sigma_{3D}$ is retrieved with, on average, a Relative Root Mean Square Error (RRMSE) < 20% and bias < 0.1% for clouds with Maximum Optical Depth (MOD) < 17, and the RRMSE of the radiances is < 0.5%, indicating very high accuracy in shallow cumulus conditions. As the MOD of the clouds increases to 80, the RRMSE and biases in $\sigma_{3D}$ worsen to 60% and -35%, respectively, and the RRMSE of the radiances reaches 16%, indicating incomplete convergence. This is expected from the increasing ill-conditioning of the inverse problem with decreasing mean-free-path predicted by RT theory and discussed in detail in Part 1. We tested retrievals that use a forward model that is better conditioned but less accurate due to more aggressive delta-M scaling. This reduces the radiance RRMSE to 9% and the bias in $\sigma_{3D}$ to -8% in clouds with MOD ~80, with no improvement in the RRMSE of $\sigma_{3D}$. This illustrates a significant sensitivity of the retrieval to the numerical configuration of the RT model which, at least in our circumstances, improves the retrieval accuracy. All of these ensemble-averaged results are robust to the inclusion of radiometric noise during the retrieval. However, individual





realizations can have large deviations of up to 18% in the mean extinction in clouds with MOD ~80, which indicates large uncertainties in the retrievals in the optically thick limit. Using the better conditioned forward model tomography can also

accurately infer optical depths (OD) in conditions spanning the majority of oceanic, cumulus fields (MOD < 80) as the retrieval provides OD with bias and RRMSE better than -8% and 36%, respectively. This is a significant improvement over retrievals using 1D RT, which have OD biases between -30% and -23% and RRMSE between 29% and 80% for the clouds used here. Prior information or other sources of information will be required to improve the RRMSE of $\sigma_{3D}$ in the optically thick limit, where the RRMSE is shown to have strong spatial structure that varies with the solar and viewing geometry.

## 1 Introduction

Remote sensing retrievals of cloud and aerosol properties are important contributors to our understanding of cloud and aerosol processes and constraining the emergent behaviour of these processes on Earth's climate (Bellouin et al., 2020; Sherwood et al., 2020). As atmospheric modelling has become more complex, there has been an increased demand for high quality observations to constrain the uncertain processes within the models and inform model development (Morrison et al., 2020).

New observational techniques are required that can provide robust statistics of small-scale, spatially resolved, cloud and aerosol microphysical parameters, so that their controlling processes can be constrained in both high- and low-resolution modelling.

In Part 1 of this study (Loveridge et al., 2022b), we described a remote sensing retrieval technique with the potential to meet these needs by providing 3D instantaneous snapshots of volumetric properties of the atmosphere at the resolution of passive

imagery. Our method uses multi-angle imagery and the Spherical Harmonics Discrete Ordinates Method (SHDOM) for modelling 3D Radiative Transfer (RT) to constrain the 3D properties of atmospheric particles, such as effective particle radius and mass concentration (Levis et al., 2020; Tzabari et al., 2022), in a process called tomography (Arridge and Schotland, 2009; Martin et al., 2014). Our retrieval algorithm for the tomography problem directly builds upon earlier work in making such retrievals tangible and computationally efficient through the use of approximate Jacobians of 3D radiative transfer that enable

the use of efficient gradient-based local optimization methods (Levis et al., 2015, 2017, 2020). We have made our method publicly available in the software package Atmospheric Tomography with 3D Radiative Transfer (AT3D) (Loveridge et al., 2022a), which was developed from the research software used in Levis et al. (2015, 2017, 2020) called pySHDOM.

Our tomographic algorithm retrieves 3D properties and makes use of 3D RT. In doing so it relaxes the twin assumptions of

independent pixels and homogeneous plane-parallel clouds that cause significant biases in operational retrievals of cloud microphysical properties (Marshak et al., 2006; Kato and Marshak, 2009; Zhang et al., 2012; Lebsock and Su, 2014; Ahn et al., 2018; Fu et al., 2019; Painemal et al., 2021; Fu et al., 2022). Another key advantage of the method is that it does not rely on expensive active remote sensing for retrieving volumetric information as in other methods (Fielding et al., 2014). Instead, the method only relies on relatively inexpensive passive imaging in the solar part of the spectrum. It thereby enables wide





swath widths and high-resolution retrievals with high Signal-to-Noise Ratio and high sensitivity to scattering particles in the size range of atmospheric clouds and aerosols (Dubovik et al., 2011; King and Vaughan, 2012; Ewald et al., 2021). These benefits position the tomographic retrieval as a means of filling the observational gap that exists for the highly heterogeneous fields of cumulus that are climatically important (Sherwood et al., 2014).

It is still unclear exactly how effective tomography will be across the range of scattering regimes present in the Earth's atmosphere. The development of tomographic retrievals in atmospheric science is at an early stage where numerical tests have yet to consider the full complexity of Earth's atmosphere and surface (Martin and Hasekamp, 2018; Levis et al., 2020; Doicu et al., 2022b; Tzabari et al., 2022). This two-part study contributes to further our understanding of the effectiveness of cloud tomography. In Part 1 of our study, we evaluated the accuracy of our approximate Jacobian for the first time and established the theory behind its effectiveness. We identified a number of issues that may occur when applying the approximate Jacobian to solve a cloud tomography problem using local optimization, particularly due to the non-linearity of the problem and the loss of sensitivity of the measurements in the diffuse-scattering limit (Levis et al., 2015; Martin and Hasekamp, 2018; Forster et al., 2021; Davis et al., 2021). Our goal here in Part 2 is to test the efficacy of our proposed retrieval algorithm for retrieving 3D volume extinction coefficient across a wide range of scattering regimes, though still in idealized conditions. For the first time, we compare cloud optical depths inferred from the tomographic retrieval against those retrieved using a 1D radiative transfer model. In Section 2, we formulate the tomography problem and review relevant past work on inverse radiative transfer, summarizing key discussions and results from Part 1 where appropriate. Section 3 presents our idealized methodology to test the efficacy of the retrieval numerically. We test our retrieval on synthetic radiances calculated from stochastically generated clouds and examine the influence of the cloud optical depth on the retrieval accuracy. We also explore the influence of using an approximate forward model and introducing radiometric noise on the retrieval accuracy. Our results are presented in Section 4 and we discuss their implications in Section 5 and highlight important areas of future work in the development of tomographic retrievals. We present our conclusions in Section 6.

## 2. Background

The objective of the tomographic retrieval, as formulated in AT3D, is to select a state vector ($\boldsymbol{a}$) that parameterizes a discrete representation of the atmospheric optical or physical properties and best fits the available measurements ($\boldsymbol{y}$) and any prior knowledge of the unknown state. The selection of the best fitting state vector is done by minimizing the scalar cost function which penalizes misfit against observations in a generalized, least-squares sense:

$$\chi^2 = \left(\boldsymbol{y} - \boldsymbol{F}(\boldsymbol{a})\right)^T \mathbf{S}_\epsilon^{-1} \left(\boldsymbol{y} - \boldsymbol{F}(\boldsymbol{a})\right) + R(\boldsymbol{a}). \tag{1}$$

In this expression, $\mathbf{S}_\epsilon$, is the error covariance matrix of the residual between the measurements ($\boldsymbol{y}$) and the forward model $\boldsymbol{F}(\boldsymbol{a})$ and accounts for both measurement uncertainty and forward model uncertainty. $R(\boldsymbol{a})$ is a differentiable regularization term





that reflects prior knowledge about the unknown state vector. The forward model $F(a)$ consists of a solution of the 3D RTE and a sampling operation that provides the forward-modelled Stokes vector at the positions and angles sampled by a sensor during the acquisition of the measurements, $y$.

The solution to the inverse problem can be stated formally as the selection of the state vector $\tilde{a}$, that minimizes the cost function subject to box constraints. The box constraints are vectors of lower bounds ($l$) and upper bounds ($u$) on each element of the state vector to ensure a valid range for physical variables. For example, liquid water content or volume extinction coefficient should be non-negative. We then have:

$$\tilde{a} = \operatorname*{argmin}_{a} \chi^2, \quad \text{s.t.} \quad l \leq a \leq u. \tag{2}$$

A local minimization method such as the Limited memory-Broyden-Fletcher-Goldfarb-Shannon method for Bounded minimization (L-BFGS-B) (Byrd et al., 1995) solves Eq. 2 for a locally optimal state vector. Figure 1 presents a flowchart of this process. To use such a method, we need to be able to compute the gradient of the cost function. The gradient of the data-fit component of the cost function is given by:

$$\frac{\partial \chi^2}{\partial a} = 2\big(y - F(a)\big)^T S_\epsilon^{-1} K, \tag{3}$$

where $K$ is the Jacobian matrix containing the partial derivatives of the $i^{th}$ output of the forward model with respect to the $j^{th}$ component of the state vector:

$$K_{ij} = \frac{\partial F_i(a)}{\partial a_j}. \tag{4}$$

We use the approximate Jacobian matrix calculation described in Part 1 to efficiently compute approximate gradients of the cost function using the approximate Jacobian $\widetilde{K}$:

$$\frac{\partial \chi^2}{\partial a} \approx 2\big(y - F(a)\big)^T S_\epsilon^{-1} \widetilde{K}. \tag{5}$$



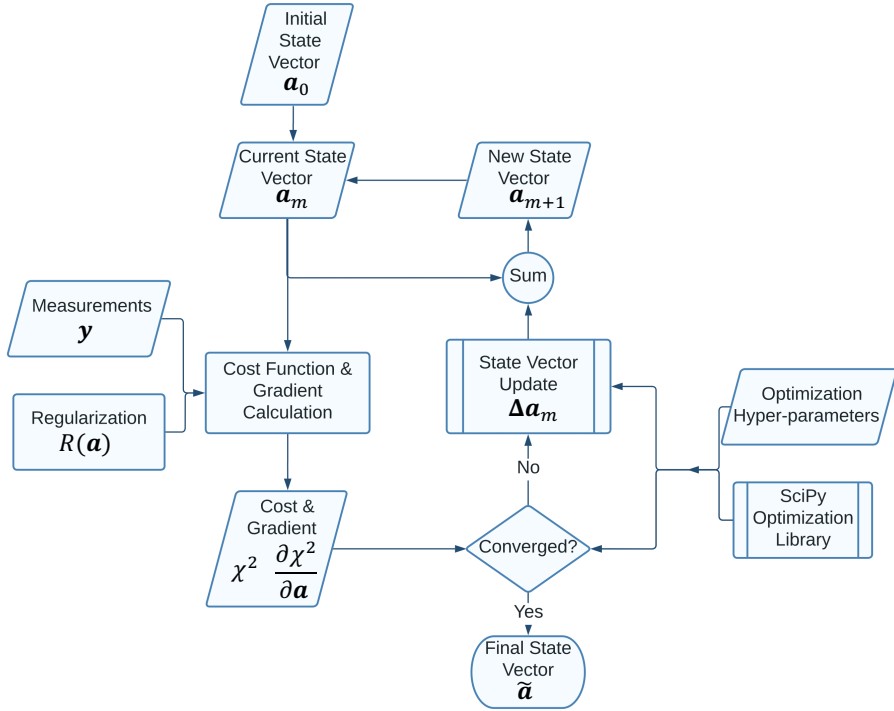

**Figure 1: A flowchart depicting the overall iterative retrieval methodology of AT3D reproduced from Part 1 (Loveridge et al. 2022). Sections 3.2 and 3.2.1 describe some of the details of how the initial state vector and optimization hyper-parameters are chosen, along with a procedure for synthetically generating some measurements using AT3D.**

Our approximate Jacobian matrix is accurate in the single-scattering limit. In this regime, it has a relative Root Mean Square Error (RMSE) of 4% with respect to finite differencing calculations (Loveridge et al., 2022b), similar to the accuracy of derivatives calculated using a forward-adjoint formulation (Doicu and Efremenko, 2019). The accuracy of the approximate Jacobian degrades as the medium becomes optically thicker, the phase function becomes more isotropic, and as the single-scatter albedo and surface albedo increase (Loveridge et al., 2022b). When the medium has single-scattering properties

representative of cloud droplets at visible wavelengths and the surface is dark (e.g., oceanic) then the relative RMSE in the approximate Jacobian reaches only 12% for finite clouds with maximum optical depths of 100.

The accuracy of the approximate Jacobian also varies with the angular resolution of the SHDOM solver when phase functions have strong forward scattering peaks (Loveridge et al., 2022b). This occurs due to the interaction of the angular resolution and

the delta-M scaling (Wiscombe, 1977) and Truncated Multiple Scattering (TMS) approximations (Nakajima and Tanaka, 1988) used in SHDOM. Lower angular resolution results in more scattering being treated as part of the direct transmission. This results in a decrease in the effective optical thickness of the medium and a corresponding increase in the accuracy of the approximate Jacobian. If the angular resolution is lowered from 16 zenith angle discrete ordinate bins to just two zenith angle



discrete ordinate bins then the error in the approximate Jacobian decreases from 12% to 8% for the cloud like media described
above.

Errors in the Jacobian calculation lead to errors in the gradient which limit the convergence of a local optimization method
such as the L-BFGS-B method. This is because, in the presence of errors, only small step sizes will accurately predict the
change in the cost function to a precision that satisfies the stability criteria in the line search of the optimization procedure
(Byrd et al., 1995; Zhu et al., 1997). If errors in the gradient are large, then the optimization may terminate far from an apparent
minimum in the cost function even when the optimization problem is linear (Shi et al., 2021).

There are several factors that complicate this retrieval methodology which uses local optimization. These factors have been
discussed in Part 1 and elsewhere in the literature. In the remainder of this section, we briefly summarize these factors which
motivate our study and its methodology. In addition to the uncertainties due to our approximate Jacobian calculation, the use
of a local optimization method on a non-linear inverse problem also introduces significant uncertainty. As the forward model
$F(a)$ is non-linear, multiple local minima in the cost function may exist. This means that the best-fitting state vector estimated
from local optimization methods may be strongly sensitive to the choice of initialization and may be far from the globally
optimal solution (Rodgers, 2000). Linearized uncertainty estimates may also be inaccurate (Rodgers, 2000; Gao et al., 2022).
So far, only local optimization methods have been employed for physics-based cloud tomography (Martin and Hasekamp,
2018; Levis et al., 2020; Doicu et al., 2022b). The proposed BFGS algorithm performs well compared to other local
optimization techniques (Doicu et al., 2022a). Global optimization methods such as ensemble-based particle filters (van
Leeuwen et al., 2019) have also been proposed for tomography in other fields (Raveendran et al., 2011). Interestingly, it has
been shown that our approximate Jacobian can outperform an "exact" linearization using forward-adjoint methods which was
suggested to be due to the ability of the approximate Jacobian to escape local minima (Doicu et al., 2022b).

The use of a local optimization method introduces the need to choose a particular initialization for each retrieval. This choice
is non-trivial, especially as clouds become optically thick. A poor choice for the spatial distribution of the extinction coefficient,
even with a correct average mean-free-path (or optical diameter), may degrade the retrieval performance. This is because the
distribution of an optically thick medium in space is highly non-linear, while adjusting the extinction coefficient within a
known volume is weakly non-linear. It has been shown that when the ground truth cloud envelope is used to initialize the
retrieval, then the performance of the retrieval improves substantially (Tzabari et al., 2022).

In addition to the complexity of selecting an initial guess appropriate for an optically thick cloud, local optimization is expected
to become much more difficult in the regions of state space where clouds are optically thick. This is due to the ill-conditioning
of the forward model (Loveridge et al., 2022b). The degree of ill-conditioning of an operator measures its instability to
inversion. The condition number of the Jacobian matrix, $\kappa(\mathbf{K})$, is the upper bound for the relative enhancement of an error in





the measurement space when it is propagated by the Jacobian matrix to the state space. We measure the magnitude of errors in a vector using the commonly used 2-norm, so then the condition number of a matrix is defined as the ratio of its largest ($s_1$)

and smallest ($s_n$) singular values:

$$\kappa(\mathbf{K}) = \frac{s_1}{s_n} .\tag{6}$$

In Part 1, we showed that the condition number increases exponentially from well-conditioned $\kappa(\mathbf{K}) \sim 10^1$ to very ill-conditioned $\kappa(\mathbf{K}) \gg 10^5$ as the optical depth of the medium increases from 0.1 to 100.0 for finite clouds. A condition number that is comparable to or larger than the inverse of the numerical precision of any floating-point numbers is effectively ill-posed

as even rounding errors can propagate to large uncertainties. The exponential behaviour of the condition number is in agreement with theoretical estimates of the stability of the continuous RT problem (Bal and Jollivet, 2008; Chen et al., 2018; Zhao and Zhong, 2019), perturbative numerical studies of radiative transfer in cloudy atmospheres (Martin and Hasekamp, 2018; Forster et al., 2021), and the field of Diffuse Optical Tomography (DOT) (Tian et al., 2010; Niu et al., 2010; Raveendran et al., 2011). This behaviour is due to the smoothing effect of multiple scattering. A smoothing operator is unstable under

inversion. This effect of radiative smoothing has been the subject of study in cloudy atmospheres (Marshak et al., 1995; Davis et al., 1997) in the context of optical depth retrieval, but is enhanced in a tomographic context due to the finer 3D discretization of the medium (Bal, 2012).

The ill-conditioning in optically thick clouds results in uncertainties that are not evenly distributed throughout the cloud. It has

been shown that measurements first lose sensitivity to changes in the cloud in the regions that are optically far from all sensors and also the sun (Niu et al., 2010; Tian et al., 2010; Forster et al., 2021; Loveridge et al., 2022b). This will result in the largest uncertainties in those regions. In Part 1, we also hypothesized that the large mismatch in sensitivities between these regions and the outer edges and illuminated sides of the optically thick clouds would cause systematic errors in retrievals that use local optimization. There is some indication that this effect occurs in practice as retrievals performed with an optically thin

initialization have underestimations of extinction in the centre of clouds, especially when they are thicker (Levis et al., 2015; Martin and Hasekamp, 2018). One focus of our work is to quantify the extent to which such systematic errors emerge in our retrievals.

In Part 1, we also identified that the magnitude of the loss of sensitivity and ill-conditioning of the forward model in the

optically thick limit can be dramatically reduced by lowering the angular resolution of the SHDOM model. This is again due to the interaction of the delta-M scaling with the angular resolution of the model due to the lowering of the effective optical depth of the radiative transfer. For the same finite clouds described above, the decrease in the magnitude of the condition number of the Jacobian matrix can reach a factor of $\sim 10^2$ when lowering the angular resolution from 16 zenith angle discrete ordinate bins to just 2 zenith angle discrete ordinate bins. This means that a lower accuracy SHDOM model is more sensitive



to the interior of an optically thick cloud and may be useful for improving retrieval accuracy despite its larger forward modelling error. This property is independent of our approximation to the Jacobian matrix (Loveridge et al., 2022b).

The objective of our study is to examine the effectiveness of the local optimization method described in Part 1 to perform cloud tomography. As part of this, we also test the sensitivity of the local optimization to the optical depth of the cloud and to
the angular accuracy used in the forward model, SHDOM. Low accuracy forward models are frequently used to accelerate the iterative solution of optimization problems (Tarvainen et al., 2009; Peherstorfer et al., 2018) and our analysis below provides a first test of whether this approach may also be beneficial in the context of cloud tomography to reduce computational cost.

## 3 Methods

We perform retrievals on synthetic clouds with stochastically generated 3D fields of volume extinction coefficient. These
synthetic clouds are designed to resemble cumuliform clouds and have maximum optical depths that range from 4 to 88. The technical details of the procedure for generating these synthetic measurements for a range of clouds across a range of optical depths is described in Section 3.1. We perform retrievals of the 3D volume extinction coefficient using perfect knowledge of the ground-truth atmosphere, surface, and cloud microphysics. This idealized configuration is sufficient to test the efficacy of the approximate Jacobian. We initialize our local optimization by assuming that the cloud is extremely optically thin, which
has been exclusively used so far in cloud tomography (Levis et al., 2015, 2017; Martin and Hasekamp, 2018; Levis et al., 2020; Doicu et al., 2022a, b). The precise methodology we use is described in Section 3.2 and the associated appendices. This configuration acts as a limiting case as the local optimization will be tested by the high degree of non-linearity of the forward model between its optically thin initialization and the, potentially optically thick, ground truth. As such, we must stress that our results are likely particular to this choice.


To isolate the fundamental limitations in using the local optimization method with the approximate Jacobian, we perform our analysis in an idealized numerical setting. We perform several "inverse crimes" in our retrievals by choosing the discretization of the retrieved medium to perfectly match the ground truth and neglecting several important sources of uncertainty such as forward model error and instrument calibration uncertainties. The first of these approximations has been routine throughout
numerical studies of cloud tomography (Levis et al., 2015, 2017; Martin and Hasekamp, 2018; Levis et al., 2020; Doicu et al., 2022a, b; Tzabari et al., 2022). Such approximations are also common in the assessment of other algorithms for atmospheric remote sensing, where synthetic measurement data are often generated using the same 1D radiative transfer model used to perform the retrievals (Delanoë and Hogan, 2008; Xu et al., 2022). Such simplifications occur despite the fact that this approximation is known to fundamentally simplify the nature of the inverse problem and thereby cause underestimates of the
true retrieval error (Rodgers, 2000; Bal, 2012). Within radiative transfer, a fixed discretization error also implies a homogeneity





assumption below a scale that is likely unrealistic for real clouds, which can cause biases in the resulting modelling of the radiative transfer (Marshak et al., 1998; Davis and Marshak, 2004; Bitterli et al., 2018).

Measurement noise tends to decrease from 1% to just 0.1% for cloudy signals as the measured radiance increases (Bruegge et al., 2002). Measurement noise has typically been included in cloud tomography studies so far (Levis et al., 2015, 2020; Tzabari et al., 2022) although in some cases, this source of uncertainty has also been neglected (Doicu et al., 2022a, b). The magnitude of this noise is much smaller than forward modelling errors that can range up to several percent in a root-mean-square sense (Evans, 1998; Cahalan et al., 2005; Pincus and Evans, 2009). These uncertainties have been neglected so far with the exception of the study of Martin et al. (2018), which used an inflated measurement error of 2% as a proxy for the modelling error.


In our study, we perform sets of retrievals that use both perfect, noise-free measurements and also those that include idealized measurement noise to ensure that our conclusions about the fidelity of the retrieval are robust to the noise free assumption. Even with this analysis, we must emphasise that our retrievals are highly simplified and so the errors in our retrievals should only be interpreted as a tentative lower bound for errors that might occur for retrievals applied to real data.


We analyse seven types of retrieval. Firstly, we have perfect model retrievals, in which we use the exact same forward model configuration and discretization of the medium during the retrieval that was used to generate the synthetic measurements. This type of retrieval uses noise free measurements and the naïve, optically thin initialization which we describe in Section 3.2. These retrievals are referred to as "Default" retrievals. Retrieval accuracy in this configuration is limited only by the non-

linearity of the inverse problem and the errors in the approximate Jacobian, both of which are deterministic.

Secondly, we perform retrievals that are the same as the "Default" except they use a low angular accuracy forward model. These are referred to as "Low" retrievals. This retrieval acts as a test of how much the retrieval changes when using an approximate forward model that is better conditioned. The differences between the "Default" and "Low" retrievals may occur

as a result of changes in the optimization trajectory either close-to or far from the local minimum in the cost function. To distinguish between these two types of effects, we perform a third set of retrievals. These retrievals use the same configuration as the "Low" retrieval except they are initialized at the ground truth and are referred to as "Low-GT". The drift in the state vector away from the ground truth during the optimization acts as a measure of the error induced by using a low angular accuracy model in the retrieval in the vicinity of the global minimum.


We also include two sets of restarted retrievals. The restarted retrievals make use of the results of the "Low" retrievals to initialize retrievals that use the perfect forward model from the "Default" retrievals. These restarted retrievals are referred to as "20th Low iteration Restart" and "Final Low iteration Restart" based on the iteration of the "Low" retrievals from which the



state vector is taken to initialize the new restarted retrieval. These retrievals test our ability to mitigate any errors forming in
the Low retrievals by using the perfect forward model.

Additionally, we perform two sets of experiments to ensure the robustness of our conclusions to the presence of noise or other
small inconsistencies between the synthetic measurements and the forward model. Firstly, we perform retrievals initialized at
the ground truth that make use of noisy measurements and the perfect model from the "Default" retrievals. These are referred
to as "Noisy-GT" retrievals. Secondly, we repeat the "Default" and "Low" experiments using noisy observations and use the
differences to assess the robustness of our results. These retrievals are referred to as "Default-Noisy" and "Low-Noisy".

**3.1 Synthetic Measurement Generation**

We generate the synthetic measurements entirely using cloud fields, RT, and instrument modelling implemented in AT3D.
The only scattering particle species under consideration in each retrieval is an isolated water cloud (i.e., no molecular scattering
or absorption). We assume that the cloud scattering is conservative ($\omega = 1$) and the phase function is from Mie calculations
of a gamma distribution of spherical water droplets with an effective radius of 10 microns and effective variance of 0.1 at a
wavelength of 0.86 microns. Vacuum horizontal boundary conditions are used and the bottom surface is prescribed to be black.
There is only a solar source, no thermal emission.

The clouds used in this study are stochastically generated rather than generated by Large Eddy Simulations, as is common in
many retrieval validation studies (Marshak et al., 2006; Kato and Marshak, 2009; Ewald et al., 2019). This choice is made
primarily for control over the media. Realistic covariances between microphysical parameters, which is one of the primary
benefits of LES (Miller et al., 2018), are not required for these retrievals as only a single spatially variable parameter is being
retrieved (i.e., volume extinction coefficient). With stochastic generation we can control the spatial variability at all scales and
produce "difficult", non-smooth extinction fields at scales of tens of meters with great ease that would otherwise require undue
computational expense with LES.

The stochastic cloud generator used to generate the extinction fields is included with AT3D. It is based on similar statistical
principles utilized elsewhere for synthetic cloud generation (Iwabuchi and Hayasaka, 2002). Such principles include the fact
that liquid water content tends to have a positively skewed distribution (e.g. lognormal) and the power spectrum of its
variability tends to follow a power law. The unique aspect of this generator is that it is targeted at generating isolated clouds
rather than large fields with periodic boundaries. The algorithm proceeds as follows:

1.  Two fields of white, gaussian noise with zero mean and unit variance are generated with skewness < 0.1 and kurtosis
< 0.5. One will form the binary volumetric cloud mask and the other will form the 3D field (e.g. extinction or liquid
        water content). This separation is done so that the smoothness of the cloud boundary can be controlled independently.





It also introduces additional small-scale variability to the field by allowing large discontinuities between in-cloud and cloud-free field values.

2. The fast Fourier transform (FFT) of each field is obtained and then scaled so that the power spectrum follows a power law with a specified exponent (e.g., -5/3). The resulting fields are then inverse Fourier transformed back into physical space.

3. The two fields are exponentiated to obtain positively skewed (e.g. lognormal) statistics typical of clouds.

4. The cloud mask field $X$ is scaled by an anisotropic Butterworth filter in physical space to enforce small values near the boundaries of the domain. This counteracts the periodic nature of the FFT process used to generate the noise and allows generation of isolated clouds that have a "blob" shape that tends to maximize in the centre of the domain. Note there is no specific requirement for a single contiguous cloud mass, though this filter does encourage it. The filtered cloud mask field $\tilde{X}$ is expressed as

$$\tilde{X} = \frac{X}{\sqrt{1+\left(\frac{R}{\alpha}\right)^8}\sqrt{1+\left(\frac{z'}{\beta}\right)^8}} \tag{7}$$

where $R$ is the horizontal distance from the centre of the domain and $z'$ is the vertical distance from the centre of the domain. The two scale parameters are set to $\alpha = 0.2$ and $\beta = 0.2$.

5. The cloud mask field is then thresholded to obtain a user specified volume cloud fraction of 10%.

6. The points in the second field are set to zero where the cloud mask field is designated as clear.

7. The second field is scaled so that its mean and variance have a user-specified vertical profile over the cloudy points and forms the 3D field of, for example, volume extinction coefficient.

We use 10 different random seeds and four different vertical profiles of horizontally-averaged extinction, thus generating 40 cloud models in four categories of maximum optical depth. Each of the four profiles prescribes a linear increase of mean extinction with height. Note that this is a more rapid increase of extinction with height than predicted by adiabatic theory, namely, extinction increases as $\sigma(h) \sim h^{2/3}$, with $h$ being the height above cloud base. At each level, the deviations from the mean are scaled so that the standard deviation is chosen to be 40% of the mean at that level. Each cloud is generated on a grid of (25 x 25 x 25) grid-points with 40 m resolution forming a domain of (1 km)$^3$. We name the four categories of cloud models, Thin, Medium, Thick, and Very Thick. They have maximum vertical optical depths of 4, 17.5, 44, and 88, respectively.

Figure 2 shows the optical depth distributions of each of the four categories of clouds (left panel) as well as a volume rendering of one of the extinction fields (right panel). Note that the decoupling of the extinction and cloud mask fields in the stochastic generator can produce true voids within the cloud. The cross sections of the extinction field (e.g., Figure 4) of a cloud reveals that the extinction fields can be much more variable than might be expected from an LES generated cloud at small scales





(Eytan et al., 2022). This is a carefully considered choice as we want to test the limits of the retrieval on non-smooth media to detect, for example, a smoothing bias.

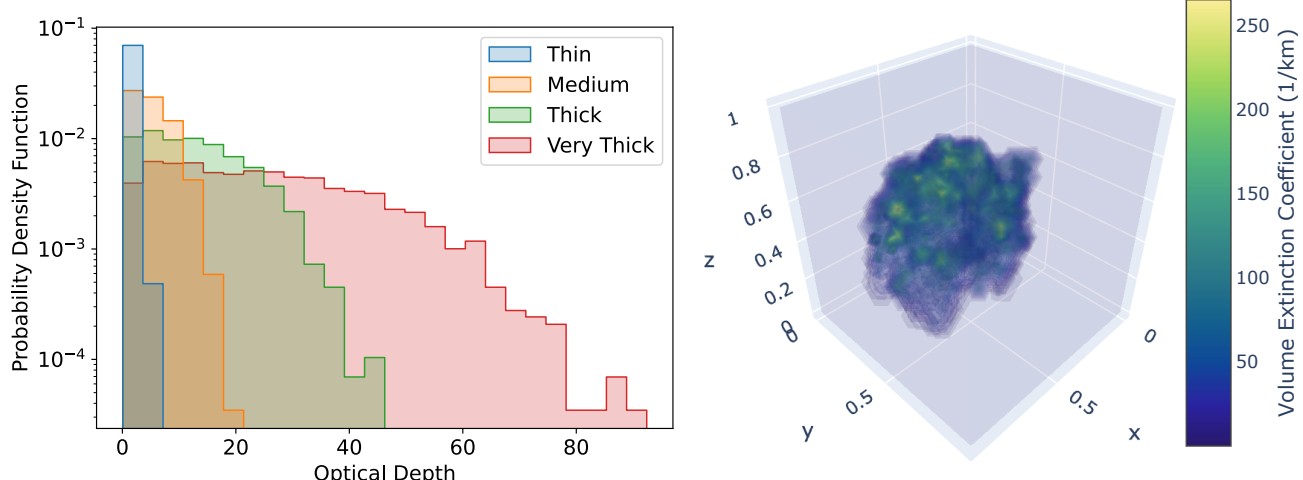

**Figure 2: (a) The probability density functions of cloud optical depth for stochastically generated clouds in the Thin, Medium, Thick and Very Thick categories. Each category contains 10 clouds. (b) A volume rendering of the 3D volume extinction coefficient of the 1st cloud realization from the Very Thick category of stochastically generated clouds. The volume rendering was created using the Plotly software.**

330

The synthetic measurements used in this study are chosen to mimic an airborne multi-angle imager such as AirMSPI operating in step-and-stare mode (Diner et al., 2013), with the exception that all measurements are acquired simultaneously in this synthetic scenario. This is a similar configuration to that which can be achieved with the upcoming Cloud-CT mission (Schilling et al., 2019), which will utilize a constellation of small satellites to obtain simultaneous multi-angle imagery. Here, we model the imaging geometry as orthographic projections of the domain with a fixed azimuthal orientation at right angles with the principal plane of the Sun, i.e., relative azimuth of 90° and –90°. The use of an orthographic projection is a highly approximate camera model, but is suitable for approximating a small domain near the centre of the swath of a push-broom sensor. In addition to a nadir view there are images with viewing zenith angles of [75.0°, 60.0°, 45.6°, 26.1°] on either side of the principal plane for a total of 9 views. This is the most used viewing angle configuration that has been used to test a 3D tomographic retrieval and has demonstrated success in several cases (Levis et al., 2015, 2017, 2020; Doicu et al., 2022a, b); so it forms a useful reference configuration on which to test the variation of retrieval accuracy with optical depth to help connect with other published results.

The solar zenith angle is chosen as 60°. This differs from other studies which have typically used a smaller solar zenith angle. We chose the larger value so that we can more easily discern the effects of any error features that are oriented with respect to the sun in our results as was hypothesised in Part 1. This choice also puts the observations close to the scattering angles of





maximum error in the approximate Jacobian (see Part 1, where we quantified the scattering angle dependence of the error in the approximate Jacobian calculation); hence a worst-case scenario.

The measurement resolution is set to 35 m. AT3D models pixel-level radiance measurements as a weighted sum of idealized
singular samplings of the radiance field within a field-of-view. For simplicity we choose a single quadrature point at each pixel center to model the pixel radiance in the simulation of the synthetic measurements. Choosing the measurement resolution (35 m) to be slightly smaller than the grid resolution (40 m) ensures that all of the grid points are evenly sampled by the measurements even with the single quadrature point per pixel. This ensures the basic numerical stability of the retrieval, by ensuring it is over constrained and that the use of regularization is not a strict requirement. The SHDOM solver uses 16 zenith
discrete ordinate bins, 32 azimuthal discrete ordinate bins, a splitting accuracy of 0.03 and a solution accuracy of $10^{-4}$ with no truncation of the spherical harmonics. The scalar approximation (no polarization) to the RTE is utilized.

In the Default-Noisy, Low-Noisy and Noisy-GT experiments, noise is included in the measurements. The noise model used for this procedure is described in Appendix A. For each of the 40 sets of measurements, a fixed set of noise perturbations are
generated through the selection of a random seed. These same noisy measurements are utilized in all retrieval experiments that make use of noisy measurements.

## 3.2 Inverse Problem Setup

The same forward model described above is used in most of the experiments except our Low retrievals. In these retrievals the angular accuracy in the SHDOM solver is reduced to just two zenith discrete ordinate bins and 4 azimuthal discrete ordinate
bins. All other numerical parameters are the same.

The state vector is chosen as a subset of the grid points that are potentially cloud containing according to a space carving procedure (Kutulakos and Seitz, 1999) that uses 2D binary cloud masks from each of the multi-angle images (Lee et al., 2018). This procedure follows Levis et al. (2020). The space carving procedure rules out cloud along the line of sight of all pixels that
are classified as clear. The details of the space carving algorithm and its performance are presented in Appendix B. The cloud masking of the multi-angle images is trivial due to the absence of a scattering surface or atmosphere (Yang and Di Girolamo, 2008). For the retrievals that use a naïve, optically thin initialization, the elements of the state vector are set as 0.01 km$^{-1}$. Additional description of the hyper-parameters of the optimization using the L-BFGS-B algorithm including the stopping conditions are in Appendix C.

**4 Results**

We now present the results of the retrievals and discuss the accuracy of the retrieved extinction fields and their robustness to noise (Section 4.1), the accuracy of inferred optical depths (Section 4.2), and the computational expense (Section 4.3). To





quantify the performance of the extinction retrieval we use the relative bias and relative RMSE of the volume extinction field, expressed respectively as:

$$\text{Relative Bias} = 100\% \times \frac{\|\sigma_{\text{retrieved}}\|_1 - \|\sigma_{\text{truth}}\|_1}{\|\sigma_{\text{truth}}\|_1},\tag{6}$$

$$\text{Relative RMSE} = 100\% \times \frac{\|\sigma_{\text{retrieved}} - \sigma_{\text{truth}}\|_2}{\|\sigma_{\text{truth}}\|_2}.\tag{7}$$

where $\sigma$ is the volume extinction coefficient. We also calculate the relative bias in the coefficient of variation (Standard Deviation divided by Mean) of the retrieved extinction field to quantify the heterogeneity of the retrieved cloud. We also apply these three metrics to the assessment of other variables such as optical depth. We refer to the error metrics in units of percent
(%) within the text for clarity. Simple differences in the error metrics between two retrievals are also reported in percent. Error metrics are defined by comparison with respect to the ground truth unless otherwise specified.

The error metrics are evaluated over all grid points in the $(1 \text{ km})^3$ domain, not the elements of the state vector, which only includes the grid points specified as cloudy by the space carving algorithm. In general, the fidelity of a 3D retrieval should be
evaluated over the physical fields in the domain rather than a metric on the state vector space. This facilitates comparison with other retrieval methods which may not use the same spatial basis. The relative Bias and relative RMSE of the extinction field are invariant to the inclusion of points for which there are no errors. The space carving approach employed here has no false negatives and as such, the values are the same as if the error metric were applied to the state vector.

**4.1 Extinction retrieval**

Let us first consider the robustness of the results to noise before interpreting them in detail. The accuracy of the Default and Low retrievals using noise free measurements are shown in Table 1. The ensemble-averaged error metrics with respect to the ground truth differ by less than 1% between the noise free retrievals (Default and Low) and the noisy retrievals (Default-Noisy and Low-Noisy). For this reason, we do not report the retrieval accuracies for the Default-Noisy and Low-Noisy retrievals. Instead, we report their differences with respect to the noise free retrievals (Table 2). Additionally, we found that the retrieval
errors of the Noisy-GT retrievals were less than 0.01% for all realizations in all cloud categories so we do not report further details of the results of those retrievals. Immediately, we can see that the ensemble-averaged results are robust to the inclusion or exclusion of noise.

The differences between the noisy and noise free retrievals increase with optical depth (Table 2) and can include large
systematic differences between retrievals for particular realizations in the Thick and Very Thick cloud categories. This latter feature can be seen in the large ranges of the error metrics in Table 2 for the Very Thick cloud category. The large deviations between noise free and noisy retrievals indicates large uncertainties in these retrievals. The differences in between the Default/Low retrievals and their corresponding noisy variants (Table 2) are much larger than the retrieval errors of the Noisy-





GT retrievals (which are negligible). This large discrepancy indicates that the issues of non-linearity and errors in the local

optimization will confound uncertainty quantification as ensemble-based uncertainty estimates will differ depending on the

choice of initialization. We revisit the implications of this point in the Discussion (Section 5). We focus the rest of our analysis

on explaining the variability in retrieval accuracy for the noise free retrievals.

**Table 1: Extinction Errors for the different retrieved clouds in each category. Means of error metrics are shown across the 10 clouds**
**in each category with the standard deviation across the 10 clouds in parentheses. See text for details of the error metrics.**

| Inversion Method | Default | | | | Low | | | | 20th Low iteration Restart | | Final Low iteration Restart | |
|---|---|---|---|---|---|---|---|---|---|---|---|---|
| Cloud Category | Thin | Medium | Thick | Very Thick | Thin | Medium | Thick | Very Thick | Thick | Very Thick | Thick | Very Thick |
| Relative RMSE (%) | 14.6 (1.7) | 19.3 (2.9) | 45.8 (7.4) | 73.0 (2.7) | 24.6 (1.5) | 37.3 (3.1) | 53.3 (4.8) | 63.2 (4.3) | 42.4 (4.4) | 62.7 (3.6) | 49.0 (6.5) | 62.7 (4.5) |
| Relative Bias (%) | 0.2 (0.1) | 0.1 (0.1) | -6.6 (6.6) | -35.6 (9.5) | 3.2 (1.4) | 5.0 (2.5) | 8.4 (6.2) | -7.5 (13.7) | -0.5 (3.3) | -14.7 (12.1) | 4.2 (5.6) | -8.3 (13.7) |
| Relative Bias in coefficient of variation (%) | -2.2 (0.5) | -3.5 (0.8) | -9.5 (4.3) | -18.4 (8.1) | -7.6 (1.4) | -9.2 (2.8) | -7.2 (2.6) | -6.0 (7.1) | -8.4 (1.8) | -8.9 (4.8) | -6.5 (2.2) | -5.8 (7.3) |

**Table 2: Differences between the extinction fields from noisy and noise free retrievals. Means of error metrics are shown across the 10 clouds in each category with the minimum and maximum across the 10 clouds in parentheses. See text for details of the error metrics.**

| Inversion Method | Default-Noisy | | | | Low-Noisy | | | |
|---|---|---|---|---|---|---|---|---|
| Cloud Category | Thin | Medium | Thick | Very Thick | Thin | Medium | Thick | Very Thick |
| Relative RMSE with respect to Noise Free retrieval (%) | 2.4 (1.9, 3.3) | 3.6 (1.5, 6.15) | 24.5 (5.4, 51.2) | 30.1 (6.3, 51.9) | 2.1 (1.5, 2.9) | 3.4 (0.8, 7.4) | 6.7 (1.7, 13.3) | 16.8 (4.2, 36.6) |
| Relative Bias with respect to Noise Free retrieval (%) | 0.0 (-0.05, 0.04) | 0.0 (-0.05, 0.04) | -3.1 (-17.5, 5.3) | 2.8 (-12.6, 16.5) | 0.0 (-0.04, 0.03) | -0.1 (-0.38, 0.1) | 0.0 (-1.7, 1.6) | -1.0 (-11.4, 8.5) |


The Default retrievals for the Thin and Medium cloud categories are uniformly accurate, with negligible bias and small relative

RMSEs (Table 1). The Low retrievals have lower accuracies than the Default retrievals for these two categories due to their

large forward modelling error. The cost function of the Low retrievals for the Thin and Medium cloud categories (Fig. 3)

begins to decrease very slowly with iteration number after only around 20 iterations and asymptotes to a larger value than for

the Default retrievals. The local optimization can proceed for ~100 iterations for the Default retrievals in the Thin and Medium



cloud categories. However, the results for the Low retrievals indicate that retrievals may converge within much fewer iterations for these clouds in the presence of modelling and instrumental errors. Given the good performance of the Default retrievals, we do not consider any Restarted retrievals for the Thin and Medium cloud retrievals. The worse performance of the Low retrieval indicates no benefit from the better conditioning and more accurate Jacobian approximation for these clouds.


The accuracy of the retrievals systematically degrades with increasing optical depth. The rate of reduction of the cost function with iteration number decreases as the optical thickness of the clouds increases (Figure 3), which can be seen by comparing the behaviour of the Medium, Thick and Very Thick cloud categories. Relative RMSE systematically increases with optical depth (Table 1) for both the Default and Low retrievals and a large bias of -36% develops in the Default retrievals in the Very

Thick category. This bias is dramatically reduced to just -7% in the corresponding Low retrievals. The convergence behaviour of the Low retrieval for the Very Thick clouds (Figure 3L) shows that the Low retrieval more rapidly increases the mean extinction with iterations particularly beyond 10 iterations, where the cloud has become optically thick. The final cost functions obtained by the Low retrievals are much lower than the Default retrieval. This is associated with a lower relative RMSE in the final modelled radiance fields of just 9% when compared to 16% for the Default retrieval. The differences between the Default

and Low retrievals for the Thick clouds are milder but are still significant, with a lower relative RMSE and slight improvements in the bias.

The dramatic improvement in the reported accuracies of the Low retrieval for the clouds in the Very Thick category indicates that there is a substantial sensitivity of the retrieval to the choice of angular resolution in the forward model. The apparent

benefit of using the Low accuracy retrieval is carried through into both types of Restarted retrievals. Using the high accuracy forward model in the Restarted retrieval does little to improve the retrieval accuracy beyond that of the Low retrieval used to initialize it (Table 1). This indicates that the local optimization algorithm is unable to further reduce the residuals despite the use of a more accurate forward model.

The faster convergence of the mean of the Low retrieval, when compared to the Default retrieval in the Very Thick category (Figure 3L) is suggestive that it is the better conditioning of the forward model that drives the improvement in the convergence. However, the improvement in the retrieval accuracy may be due to compensating biases in the approximate forward model. To help us distinguish between these effects we make use of the results of the Low-GT retrievals (Table 3). These retrievals are initialized with the ground truth cloud, but use the Low accuracy forward model. The resulting retrieval errors are therefore

a local estimate of the error in the retrieved state due to forward modelling errors in the vicinity of the ground truth. The retrieval error for the Low-GT Thin and Medium clouds are similar to those of the standard Low retrieval. This further solidifies the good behaviour of the retrievals in these cloud categories.







**Figure 3: Retrieval performance for the Default (solid lines) and Low (dashed lines) retrievals as a function of iteration number for clouds in the Thin (A, B, C), Medium (D, E, F), Thick (G, H, I) and Very Thick (J, K, L) categories. Each coloured curve corresponds to a different cloud realization. The first column (A, D, G, J) shows the cost function normalized by its initial value. The second column (B, E, H, K) shows the relative RMSE (Eq. 6) in the retrieved volume extinction coefficients. The third column (C, F, I, L) shows the relative bias (Eq. 7) in the retrieved volume extinction coefficient. See main text for details of the error metrics. Note the logarithmic scale of the iteration number, given that computational expense is linear in iteration number.**

**Table 3: Extinction Errors for the Low-GT retrievals. Means of error metrics are shown across the 10 clouds in each category with the standard deviation across the 10 clouds in parentheses. See text for details of the error metrics.**

| Inversion Method | Low-GT | | | |
|---|---|---|---|---|
| Cloud Category | Thin | Medium | Thick | Very Thick |
| Relative RMSE (%) | **15.6** (1.6) | **27.7** (2.8) | **38.1** (8.0) | **18.6** (6.4) |
| Relative Bias (%) | **3.1** (1.4) | **5.0** (2.5) | **10.2** (5.2) | **6.2** (4.4) |
| Relative RMSE with respect to the Low retrieval. (%) | **17.8** (1.8) | **22.2** (2.2) | **34.8** (4.1) | **68.6** (12.6) |
| Relative Bias with respect to the Low retrieval (%) | **-0.1** (0.0) | **-0.1** (0.2) | **1.8** (1.8) | **16.9** (14.7) |

The relative bias of the Low-GT retrieval for the Thick cloud category is 10.2% in the ensemble average (Table 3). This shows that much of the difference between the relative Bias in the Default and Low retrievals can be explained by the forward model error. In particular, the Low accuracy model produces smaller radiances for a given optical thickness of the cloud, so the cloud must be biased high in optical thickness to minimize misfit against the measured radiances. The retrieval bias of the Low-GT retrieval in the Very Thick clouds is only 6.2%. This is much smaller than the difference in relative bias between the Default retrieval and the Low retrieval. As such, forward model error in the vicinity of the solution cannot entirely explain the discrepancy between the two techniques. Part of this discrepancy is caused by the fact that the Default retrieval does not converge to the vicinity of the truth in terms of average extinction and so the forward model error at the ground truth is not representative. However, the bias difference is still much larger than the 10.2% bias that occurs for the Thick category. The other component of the difference between the Default and Low retrievals is because the trajectory of the local optimization from the optically thin initialization to the final retrieved state is very different.

To better understand the differences in the optimization trajectories between retrievals that use different forward models let us examine the spatial structure of the extinction errors. Understanding the character of these errors is important for us to understand how to mitigate these errors, particularly in the optically thick limit and whether a low accuracy, better conditioned





forward model may be a part of a mitigation strategy. In doing so, we explain how such a biased extinction field can be
retrieved that still provides a relatively small misfit against the radiance measurements in the optically thick limit.

As a starting point in our analysis of the spatial structure of the retrieval errors let us note that all retrievals have underestimated
the variability in the retrieved extinction field, especially at larger optical depths in the Default retrieval. This is demonstrated
through the errors in the coefficient of variation of the extinction field (Table 1). We further examine the spatial dependence
of this error by examining cross sections of the retrieved extinction field. In Figures 4 to 7 we show cross-sections of the
volume extinction coefficient for cloud realization #4 of the stochastic clouds as it is representative in providing all of the key
points from the other realizations when visualizing the spatial structure of the errors. We show just the Default and Low
retrievals as the Restarted retrievals are qualitatively similar to the Low retrievals. Cross sections of the extinction field are
shown for the slices through the centre of the domain aligned along the azimuthal directions of the sensors and the sun. We
refer to these slices as the measurement plane and solar plane, respectively.

The high-frequency details of the extinction field are retrieved almost perfectly for the Thin (Fig. 4) and Medium (Fig. 5)
clouds, with only mild errors apparent in the measurement plane of the Medium cloud. On the other hand, for the Thick (Fig.
6) and Very Thick (Fig. 7) clouds, the extinction field is extremely smooth. There is a gradient in extinction in the solar plane
from the illuminated side (left) to the shaded side (right) of the retrievals of the Thick and Very Thick clouds. Fine details of
the extinction field are retrieved on the illuminated side, but not on the shadowed side. Additionally, the largest values of the
retrieved extinction tend to be at the edge of the cloudy volume (as defined by space carving) in the measurement plane with
the smallest values in the centre. This feature is most apparent in the Default retrievals.

To examine the scale-by-scale apportionment of the error across the whole ensemble of retrievals, we use Fourier analysis.
Fourier analysis is a global analysis so it cannot identify, for example, a lower error in small-scale features on the illuminated
side of the cloud. To compress the information, we show just the isotropic, ensemble averaged power spectral density of the
retrieved extinction fields (Fig. 8) and their errors (Fig. 9). The retrievals tend to be significantly smoother than the ground
truth at small scales. The scale below which the retrieval has less variability than the truth increases from 125 m to 250 m as
the clouds become optically thicker. Note that the default retrieval for the Very Thick clouds tends to underestimate variability
at all scales, an error that is reduced in the Low and Restarted retrievals consistent with the reduction in the underestimate of
variability (Table 1). Note that the reproduction of approximately correct power spectra at large scales doesn't necessarily
indicate that the retrievals are error free at these scales, since the phasing of the spatial modes can still disagree and lead to
error proportional to the amplitude of the mode.

The errors in the Default retrievals of the extinction for the Thin and Medium cloud categories are actually band-limited to
small scales (Fig. 9). Errors tend to occur primarily at larger scales for the other retrievals, especially in the Thick and Very





Thick categories. This indicates that even large-scale features are not accurately retrieved in the Thick and Very Thick clouds, in general, in concurrence with Fig. 6 and Fig. 7. The primary benefits of the Low and Restarted retrievals in terms of overall

reduction of RMSE are felt at the largest spatial scales (Fig. 9).

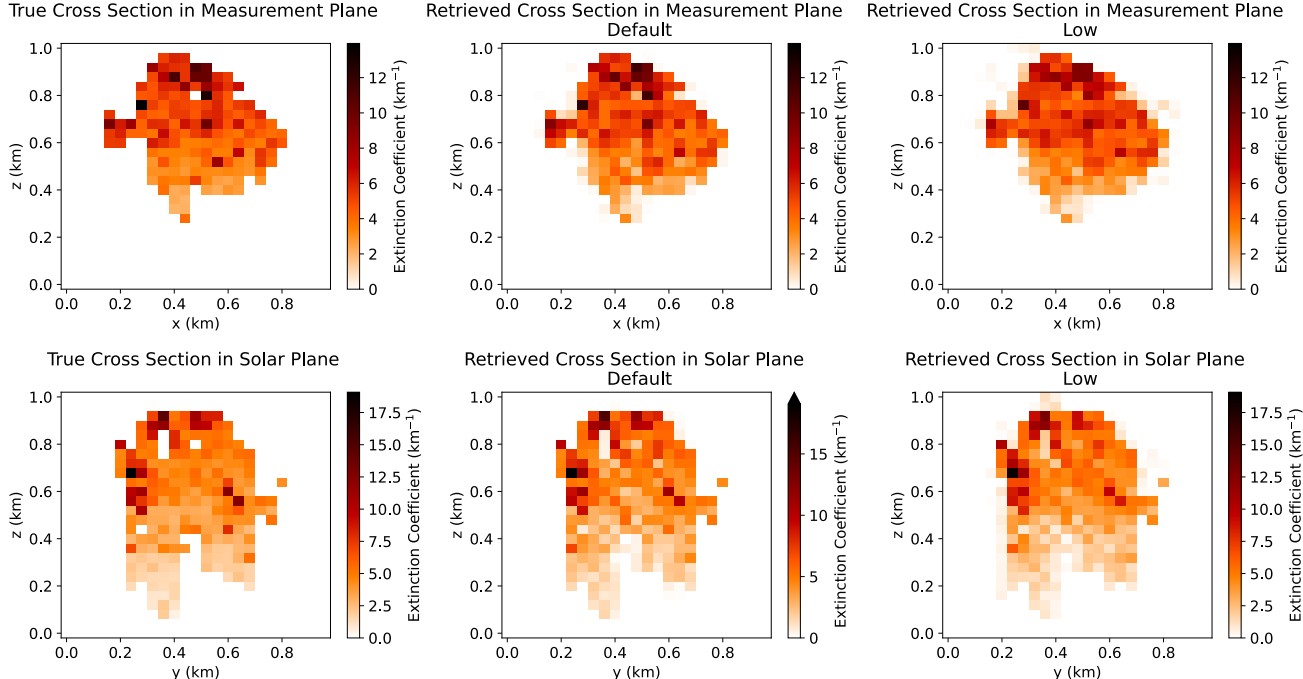

**Figure 4: Vertical cross sections of the true extinction field (Left), Default retrieved extinction field (Middle) and the Low retrieved extinction field (Right) for Cloud #4 in the Thin category. Cross sections are in the y = 0.48 km plane (Top) and the x = 0.48 km plane (Bottom). The sun points in the positive y direction, while the viewing directions are aligned along the x axis.**





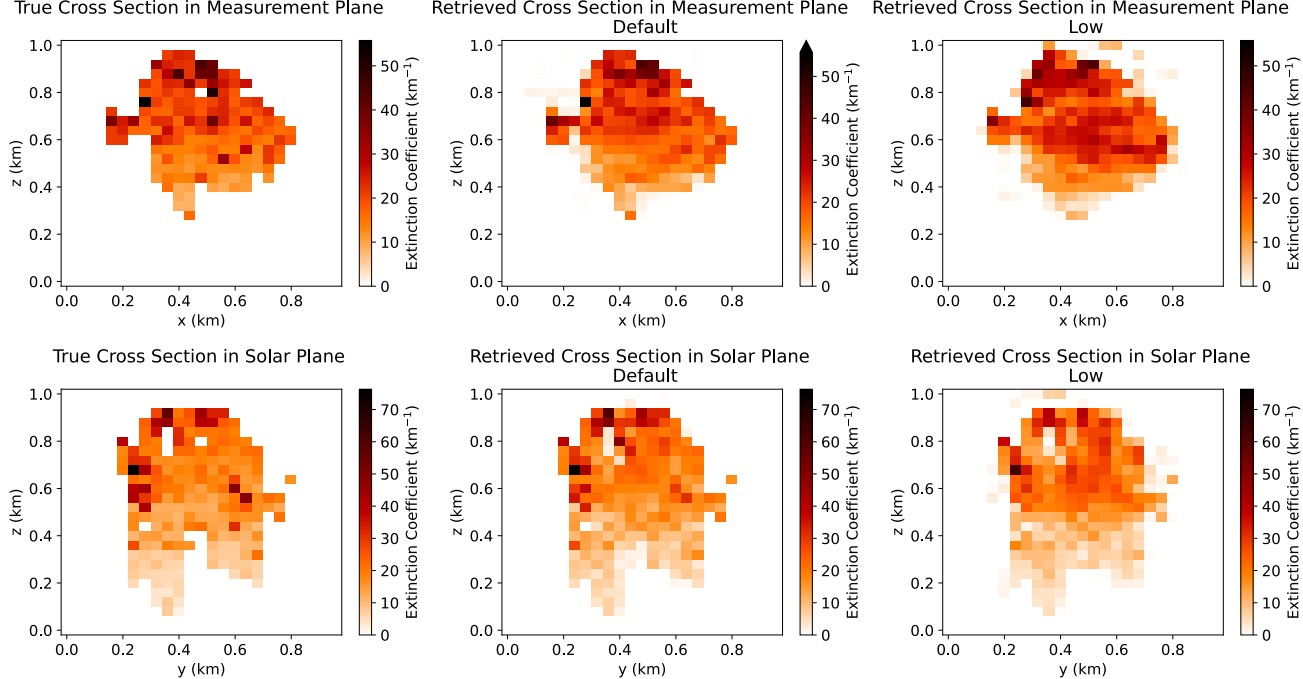


Figure 5: As in Figure 4 but for the **Medium cloud category.**

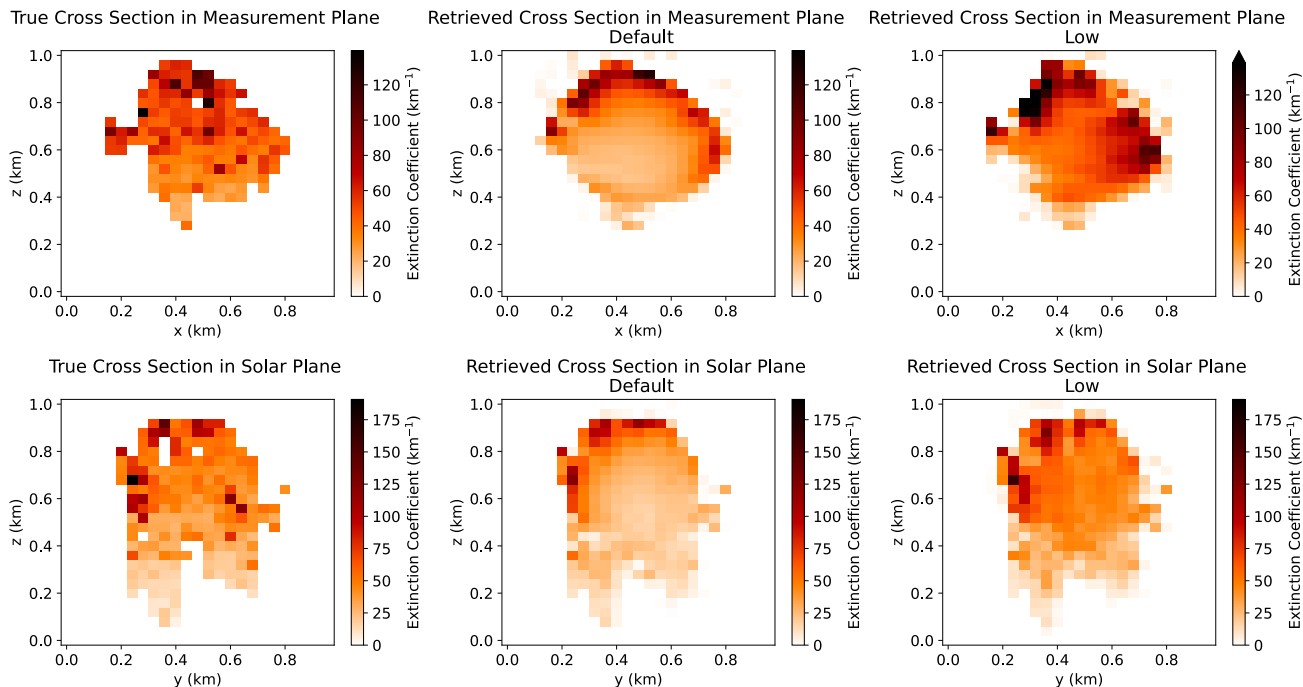

Figure 6: As in Figure 4 but for the **Thick cloud category.**



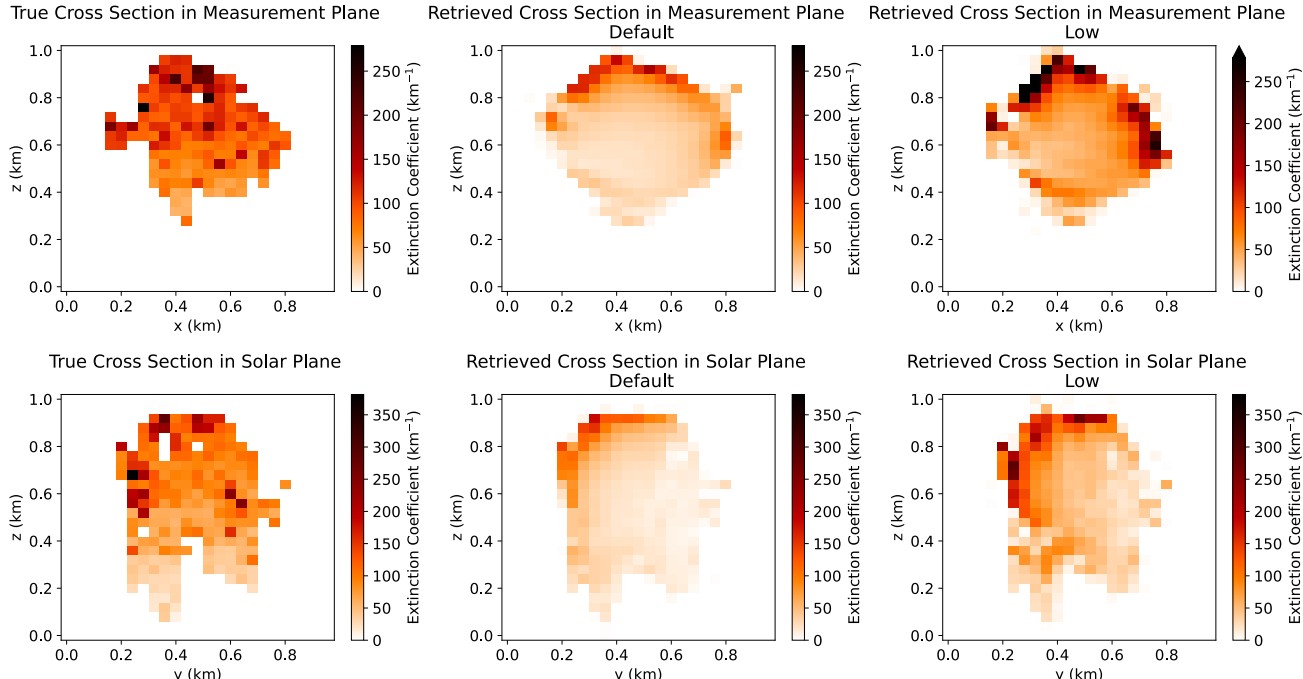


**Figure 7: As in Figure 4 but for the Very Thick cloud category.**





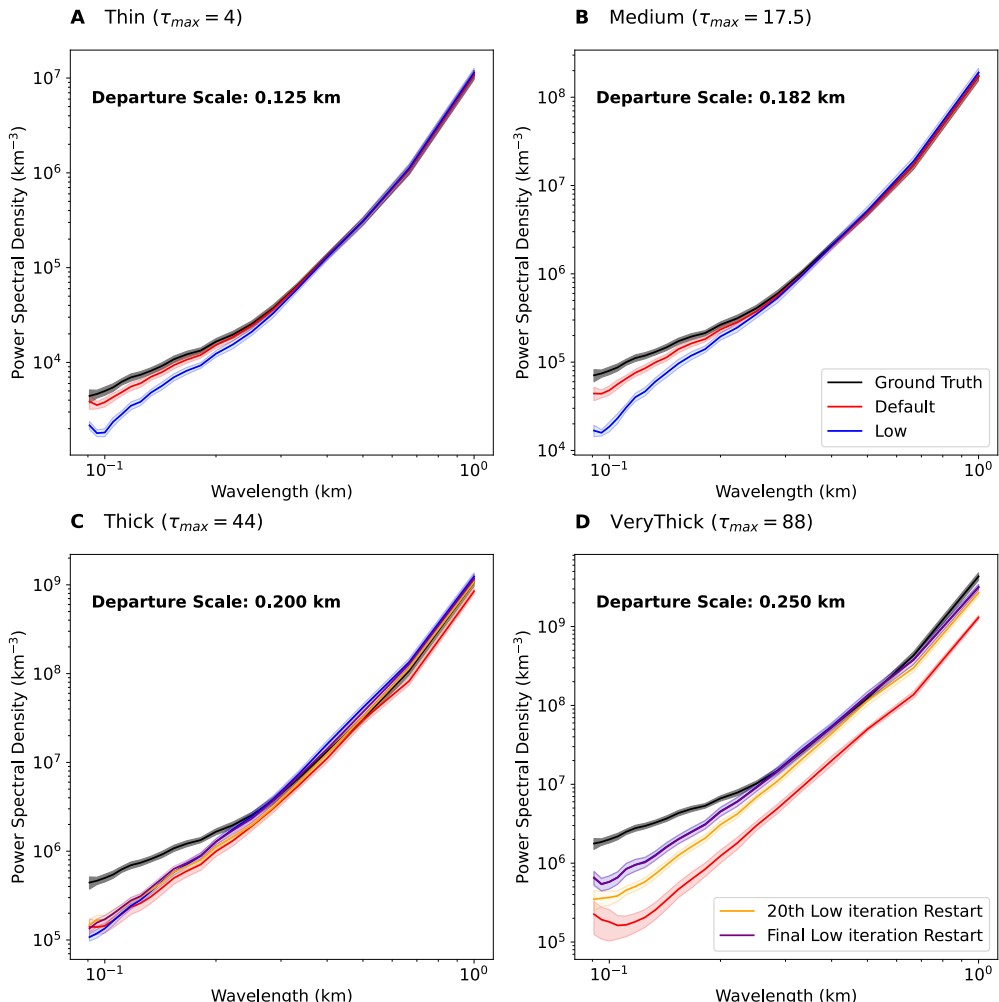

**Figure 8:** The ensemble averaged, isotropically averaged power spectral density of the volume extinction coefficient for the Thin (A), Medium (B), Thick (C) and Very Thick (D) cloud categories. Solid lines are the ensemble averages, and the shading indicates +/- one standard error in the mean. The spatial scales of the first departure from the ground truth spectrum by the best performing retrievals for each cloud category are marked in the top left corners of each figure. Note in panel D, the Low and Restarted Final curves are indistinguishable.

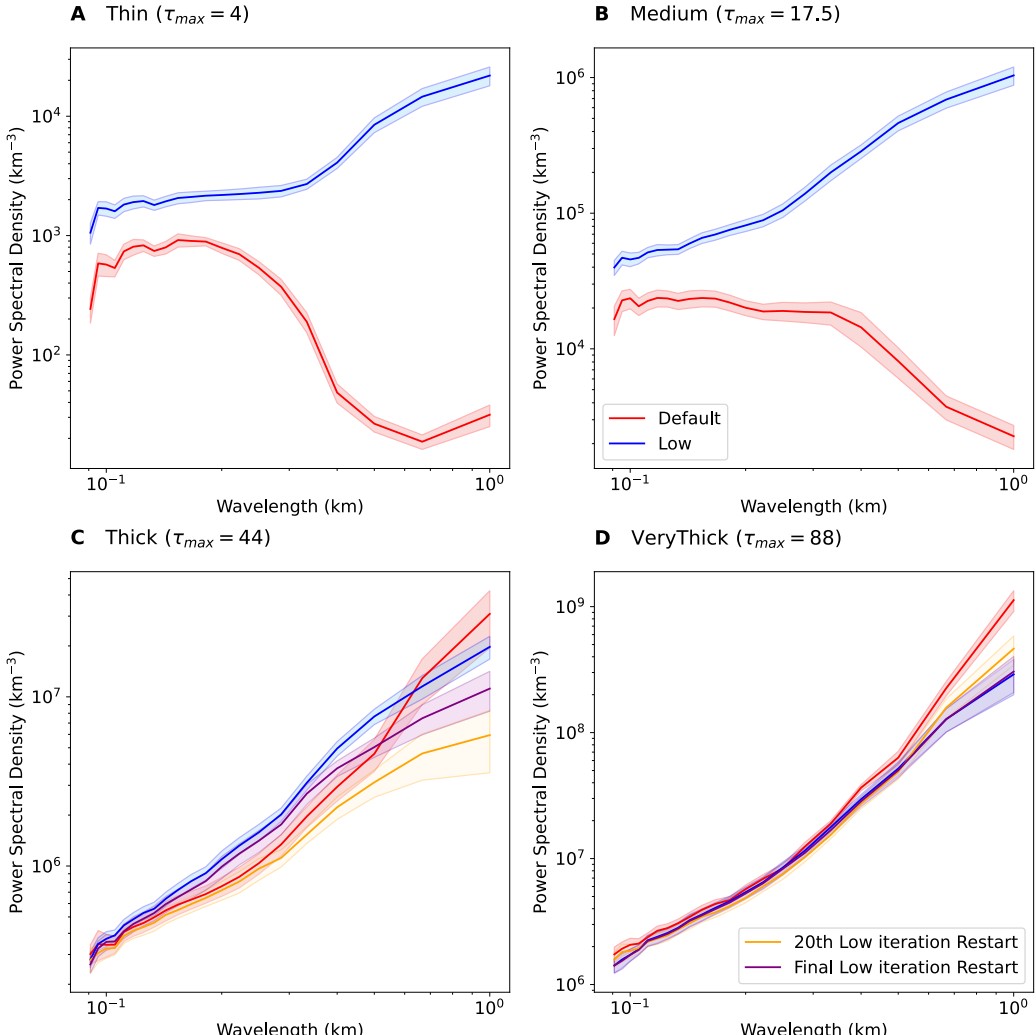

**Figure 9: The ensemble averaged, isotropically averaged power spectral density of the errors in the volume extinction coefficient for different retrievals of the clouds in the Thin (A), Medium (B), Thick (C) and Very Thick (D) categories. Solid lines are the ensemble averages, and the shading indicates +/- one standard error in the mean. Note in panel D, the Low and Restarted Final curves are indistinguishable.**

The spatial structure of the retrieval error for the Default and Low retrievals of the clouds in the Thick and Very Thick categories for the naïve, optically thin initialization, give us some important insight into some of the issues that arise in retrievals of optically thick clouds. The details of the spatial structure of the error in our retrievals are unique to our choice of initialization. In the discussion below, we distinguish between the explanation of our retrieval errors and hypotheses that we make about the general behaviour of the local optimization in the optically thick limit.





For our particular retrievals, the optically thick cloud state is approached from an optically thin cloud state. During the early
iterations, the residuals are highly smooth as we initialize with an extremely optically thin cloud. As such, the optimization
updates the cloud by reducing the bias in the radiances and fills the cloud with a relatively smooth extinction field. As the bias
reduces, the cloud becomes optically thick and now the optimal strategy to reduce the cost function is to further increase the
extinction on the illuminated side of the cloud and at the cloud edge in the plane of the measurements. This patterning is
because the magnitudes of the Jacobian elements are largest in these positions, as described in detail in Part 1. With the bias
in the cost function reduced, the radiance residuals can be multi-signed and mapped more strongly to smaller-scale spatial
features in the extinction field. However, the cloud is now optically thick and this mapping is highly ill-conditioned. So, it is
difficult from a linearized perspective to redistribute extinction between the outer and inner portions of the cloud while
avoiding an increase in the cost function. Tiny step sizes must be utilized to avoid instability, but even these are corrupted by
the increasing error in the approximate linearization. Eventually, cost function reductions are so small that the stopping
condition of the optimization is achieved. This results in the presence of gradients in cloud extinction from illuminated to
shadowed side and from cloud edge to cloud core in the retrieved state.

Based on the behaviour of the retrievals, we argue that the improvement of the Low retrieval relative to the Default retrieval
for these Very Thick clouds is driven by the better conditioning of the forward model. The ratio of the magnitude of the
Jacobian elements tends to be more similar between cloud edge and cloud centre, and the illuminated and shaded sides of the
cloud for the better conditioned low accuracy forward model (Loveridge et al., 2022b). The result of this is that the extinction
is distributed more evenly throughout the cloud during the increase in the extinction coefficient even at larger optical depths.

The smoothness of the extinction field in our retrievals (e.g. the coefficient of variation in Table 1) arises because of the
smoothness of the initial residuals and the smoothing effect of the adjoint operator that is used to calculate the gradient. The
solution operator of the radiative transfer is smoothing and so is its adjoint. In the ill-conditioned limit, local optimization tends
to converge slowly while remaining smooth. This contrasts with direct inversion methods that can produce wild irregularities
in the solution when the problem is ill conditioned. The greater smoothness of the Default retrieval for the Very Thick clouds,
compared to the Low retrieval, can be seen as a symptom of the stronger ill-conditioning that occurs in the Default retrieval
before details in the extinction field are retrieved.

The smooth extinction fields that we retrieved in the optically thick limit that are biased low are only one of a set of
configurations of clouds that will produce similar consistency in radiances against the measurements. Compensating errors in
the radiance field occur between biases in the mean and the variability of a retrieved extinction field. This is because a low
bias in extinction increases the mean-free-path, hence enhancing transmission, while decreasing spatial variability decreases
the mean-free-path, hence decreasing transmission (Cairns et al., 2000; Davis and Marshak, 2004; Forster et al., 2021). These



compensating errors can be larger in optically thick clouds without substantially affecting the outgoing radiance field due to the strong smoothing process of scattering (Davis et al., 2021).

We must emphasise that our results are not evidence of a radiative smoothing bias in tomographic retrievals similar to the one identified in the IPA. The resolving power of the measurements is affected by not just the spatial resolution but also the angular spacing of the measurements. In the optically thick limit, there is an ambiguity in the state when using only radiance measurements in the retrieval. The smoothness of our retrieved extinction fields is an artefact of our choice of retrieval algorithm (and particularly its initialization). Retrievals with other initializations may encounter the opposite form of

compensating error. For example, a retrieval may be initialized optically thick and highly variable, which results in the retrieved cloud being too optically thick and too variable. At this point, this is a hypothesis about the general behaviour of retrievals in the optically thick limit and remains to be tested.

### 4.2 Inferred optical depths

In addition to the analysis of the volume extinction coefficient field, it is also helpful to consider how the retrieval performs

for inferring optical and radiative properties such as cloud optical depth. Cloud optical depth is a widely used single variable for describing the optical properties of cloud and, for this variable, we can make a direct comparison with the performance of operational retrievals using 1D RT.

Table 4 shows the retrieval errors in optical depth distributions from the different tomographic retrievals. Retrieval errors in

optical depth are substantially smaller than in the extinction field, reflecting the relatively uncorrelated spatial distribution of extinction errors. For comparison, Table 5 displays retrieval errors from an optical depth retrieval using only nadir radiance and the assumptions of the IPA, that is, using homogeneous plane-parallel cloud models and 1D RT to generate look-up tables (LUTs) of optical depth as a function of measured radiance. Errors are substantial for the IPA retrieval, even with radiances at the same resolution as the optical depth so that sub-pixel heterogeneity effects are absent. The IPA retrieval outperforms the

default tomographic retrieval of the Very Thick clouds, but the margin is well within the ensemble standard deviations of both techniques. Moreover, even accounting for the ensemble variability, the IPA is significantly out-performed by both Restarted retrievals, which have small radiance residuals. The tomographic retrieval also outperforms the IPA retrieval when inferring the coefficient of variation of the optical depth distribution as the IPA retrieval tends to have a bias in the estimate of the distribution width that changes sign with increasing optical depth (Iwabuchi and Hayasaka, 2002). Clearly, the tomographic

retrieval demonstrated here is much more appropriate for inferring cloud optical depths in cumuliform clouds than IPA-based retrievals, even without considering the substantial benefits of retrieving the 3D distributed extinction field.





**Table 4: Optical Depth Errors for different tomographic retrievals. Means of error metrics are shown across the 10 clouds in each category with the standard deviation across the 10 clouds in parentheses. See text for details of the error metrics.**

| Inversion Method | Default | | | | Low | | | | 20th Low Iteration Restart | | Final Low Iteration Restart | |
|---|---|---|---|---|---|---|---|---|---|---|---|---|
| Cloud Category | Thin | Medium | Thick | Very Thick | Thin | Medium | Thick | Very Thick | Thick | Very Thick | Thick | Very Thick |
| Relative RMSE (%) | **2.8** (0.5) | **7.2** (1.6) | **24.1** (6.1) | **55.6** (4.1) | **8.1** (1.2) | **17.7** (3.2) | **28.5** (5.4) | **36.5** (5.0) | **19.9** (2.2) | **38.8** (4.7) | **24.5** (5.9) | **36.4** (5.4) |
| Relative Bias (%) | **0.2** (0.1) | **0.0** (0.1) | **-6.6** (6.6) | **-35.6** (9.5) | **3.2** (1.4) | **5.0** (2.5) | **8.3** (6.1) | **-7.6** (13.7) | **-0.5** (3.3) | **-14.7** (12.1) | **4.1** (5.6) | **-8.3** (13.7) |
| Relative Bias in coefficient of variation (%) | **-0.1** (0.1) | **-0.7** (0.4) | **-4.8** (1.3) | **-14.0** (6.0) | **-2.6** (0.9) | **-1.6** (1.6) | **0.1** (2.8) | **-3.2** (3.0) | **-1.9** (1.2) | **-5.0** (3.7) | **-0.3** (2.3) | **-3.3** (3.3) |

**Table 5: Optical depth errors for a Look Up Table (LUT) based retrieval of optical depth from the nadir radiance using 1D radiative transfer. Means of error metrics are shown across the 10 clouds in each category with the standard deviation across the 10 clouds in parentheses. See text for details of the error metrics. The LUT uses the ground truth microphysics and is made of 20 points with linear spacing between 0 and 5 optical depths and 40 points with logarithmic spacing between 5 and 100. A cubic interpolation is used. The best fitting optical depth is chosen using the L-BFGS-B routine. The angular accuracy used to form the LUT is the same 625 as the ground truth 3D simulations but a stricter splitting accuracy of $10^{-3}$ is used.**

| Inversion Method | IPA / LUT | | | |
|---|---|---|---|---|
| Cloud Category | Thin | Medium | Thick | Very Thick |
| Relative RMSE (%) | **29.4** (2.7) | **38.9** (4.8) | **45.7** (6.2) | **78.7** (14.1) |
| Relative Bias (%) | **-23.2** (2.7) | **-31.9** (4.5) | **-31.9** (10.3) | **-16.9** (25.0) |
| Relative Bias in coefficient of variation (%) | **-7.4** (1.8) | **-7.0** (3.4) | **1.7** (7.9) | **31.2** (15.7) |

**4.3 Computational expense**

As always, it is important to also consider the computational expense of a retrieval method alongside its performance when evaluating its utility. Table 6 evaluates the computational expense of the retrievals. It shows that the use of the low accuracy 630 SHDOM solver in the Low retrieval almost eliminates the contribution of the RTE solver to the retrieval time, which is instead dominated by the approximate computation of the gradient and radiance calculation. This latter calculation is still unoptimized, and its cost is further discussed in Appendix F of Part 1 (Loveridge et al., 2022b). The gradient calculation is easily parallelized using multi-threading as each observable can be evaluated independently. In our case, we parallelize using 4 threads. True



CPU time is therefore roughly four times larger than wall-time for this portion of the computational cost of each iteration.

Computational expense is, naturally, significantly larger for the tomographic retrieval than for the IPA-based retrieval. For comparison, the tomographic retrieval is, very roughly, two orders of magnitude more expensive than aerosol retrievals using multi-angle radiances and the IPA assumptions with online RTE calculations, neglecting differences in hardware (Gao et al., 2021).

**Table 6: Computational Expense of the Retrievals. Means across the ten clouds are shown, with standard deviations across the ten clouds in parentheses. Average cumulative Total CPU time is essentially linear in iteration number so the computational expense vs. accuracy trade-off of terminating the retrievals earlier can then be read from this Table and Figure 3. All computations were performed on a 2.3 GHz Intel Core i5.**

| Inversion Method | Default | | | | Low | | | |
| --- | --- | --- | --- | --- | --- | --- | --- | --- |
| Cloud Category | Thin | Medium | Thick | Very Thick | Thin | Medium | Thick | Very Thick |
| Total CPU Time per iteration (s) | 43.6 (7.3) | 53.2 (10.9) | 85.8 (8.5) | 115.9 (48.8) | 29.6 (6.1) | 31.6 (10.0) | 36.3 (13.1) | 30.2 (6.5) |
| Percentage of Total CPU time spent on SHDOM solution (%) | 20.4 (1.6) | 43.7 (5.2) | 54.5 (6.6) | 57.0 (7.1) | 0.5 (0.1) | 1.2 (0.2) | 2.3 (0.2) | 3.6 (0.6) |
| Average number of Objective Function Calls per iteration | 1.60 (0.27) | 1.17 (0.15) | 1.30 (0.27) | 1.50 (0.58) | 1.58 (0.39) | 1.82 (0.56) | 1.83 (0.69) | 1.47 (0.29) |

Several acceleration methods are possible. The computational scaling of the SHDOM solver to larger domains using MPI-based parallelization is documented in Pincus and Evans (2009). The solver CPU time may be decreased by using the SHDOM solution from the previous optimization iteration to initialize the next one. We utilized this feature in Part 1 to accelerate finite differencing calculations, though this method may not be stable to large changes in cloud between iterations, particularly when the adaptive grid is utilized. A more sophisticated version of this principle is to utilize PDE-constrained optimization which

jointly refines both the inverse problem and the RTE solution at the same time, leading to order of magnitude decreases in computational expense (Abdoulaev et al., 2005). The use of low accuracy RTE solutions far from the solution appears valuable, through their ability to reduce the computational expense of the solver and increasing the convergence rate of the optimization.

The RTE solver cost may be mitigated in operational applications through the development of a statistical emulator for the

SHDOM RTE solution similar to Gao et al. (2021), though this work should likely proceed only once the retrieval algorithm is at a greater level of maturity. The convergence rate of the retrieval, and hence total computational expense, and final accuracy may also be improved through the development of preconditioning strategies. The problem of ill-conditioning in the transition to the diffuse regime has been widely recognized in Diffuse Optical Tomography and some mitigation strategies have been





proposed (Tian et al., 2010; Niu et al., 2010). Extension of this work for a nonlinear preconditioning transform (De Sterck and
Howse, 2018) may be effective for our application. In our view, the computational expense of the tomography retrieval is not
an oppressive limitation of the technique. Significant computational resources are routinely utilized simply to simulate
analogues of the atmospheric system. The technique only calls for similar resources to be directed to study the atmosphere as
it really is.

## 5 Discussion

The good performance of the retrievals of the clouds in the Thin and Medium categories is highly encouraging, as is their
robustness to noise and their robustness to forward model error in this idealized scenario. Clearly, in these clouds, the
approximate Jacobian is not a limiting factor in the local optimization, and the local optimization itself is a highly efficient
method for performing the retrieval. While we must be careful not to extrapolate quantitative performance from our idealized
scenario to real world conditions, we can confidently conclude that the performance of the tomography for isolated clouds
within this scattering regime will be limited by forward modelling error and instrumental errors, rather than the difficulty of
the inverse problem. Given that many trade cumulus tend to be smaller than 800 m in geometric depth (Guillaume et al., 2018;
Chazette et al., 2020) and that the average adiabatic fractions for these clouds can be significantly less than unity (Eytan et al.,
2022), many of these clouds, particularly in clean-conditions will confidently have maximum optical depths less than 40.

A minimal amount of prior information will be required to perform retrievals of these optically thinner shallow cumulus. In
fact, our results cement the conclusion that small heterogeneous clouds are actually the clouds for which remote sensing using
passive imaging is easiest, counter to the paradigm of 1D radiative transfer. This means that the radiance measurements
themselves will provide a large amount of information that is independent of the models that might be used to form any priors
such as LES. This is hugely encouraging as the development of cloud tomography techniques is motivated in part by the need
to develop measurements that provide independent constraints on the behaviour of LES models (Morrison et al., 2020). This
fact solidifies the strengths of physics-based retrievals in comparison statistical methods. While statistical methods trained
using LES data are also attractive options to perform retrievals using 3D radiative transfer(Nataraja et al., 2022; Ronen et al.,
2022), they lack a ground truth training data set and are not independent of the LES that they may be used to evaluate.

One of the key issues that might complicate our extrapolation of tomography performance to real clouds is the potential for
cloud organization to obscure oblique views and thereby lower retrieval accuracy. Additionally, there is the issue of how the
retrieval accuracy depends on the treatment of the horizontal boundary conditions and domain size. These issues have yet to
be investigated systematically, requiring much larger domains, hence computational expense, and therefore MPI-based
parallelization, which is not yet implemented in AT3D. Observed statistics of cloud-to-cloud separation suggest that trade
cumulus are tightly spaced relative to their horizontal size, and therefore thickness (Zhao and Di Girolamo, 2007). Early
numerical results of radiative transfer with idealized cloud geometries show that the radiative effects of cloud-cloud





interactions are significant in this regime (Weinman and Harshvardhan, 1982; Schmetz, 1984; Kobayashi, 1988). As such, investigating the sensitivity of the retrieval accuracy to cloud-cloud interactions, particularly the horizontal boundary conditions, is a high priority for establishing the efficacy of cloud tomography in more realistic conditions.


Given the success of the local optimization and approximate Jacobian in the optically thin limit, the proposed retrieval is also applicable to other optically thin scattering media such as cirrus or aerosol. For cirrus clouds, the primary barrier will likely be whether the clouds have sufficient horizontal variability for the multi-angle measurements to constrain their vertical structure. Horizontal homogeneity increases ill-conditioning, introducing ambiguity in the retrieved extinction fields (Martin

and Hasekamp, 2018), though this effect is relatively minor in the optically thin limit (Loveridge et al., 2022b). Additionally, the more complex microphysical characteristics of ice and aerosols, where the shape and composition are not well known a priori, raises the question of how effective microphysical retrievals for these scatterers will be. In contrast, tomographic retrievals of liquid cloud microphysics have already been demonstrated (Levis et al., 2020) and promise to be effective in the optically thinner cumuliform clouds discussed above.


In the optically thick limit, many issues have been identified. We have provided further quantification of the degradation of retrieval accuracy and convergence in the optically thick limit due to ill-conditioning that had been previously identified (Levis et al., 2015; Martin and Hasekamp, 2018). Retrieval uncertainties are clearly large in the optically thick limit, as indicated by the significant divergences of the retrieved extinction field due to small perturbations to the measurements. However, even

these perturbative estimates are tied to our choice of initialization and are not globally representative (Rodgers, 2000; Gao et al., 2022). We showed that while we can easily diagnose retrieval accuracy using the ground truth, we currently have no effective means to predict retrieval uncertainties. Developing a computationally efficient uncertainty model is a critical step in the development of the tomographic retrieval. Particle filter methods have the potential to address this need (Hu and van Leeuwen, 2021), though work is needed to verify their applicability to the tomography problem and optimize them for this

application.

While global optimization methods are comprehensive, they are also much more computationally expensive than the local optimization method employed here with AT3D. Minimizing the amount of time spent in global optimization through the improvement of the local optimization is always beneficial. Correspondingly, developing more effective initialization

strategies and other acceleration methods will improve the retrieval problem. Methods to estimate the order of magnitude of the optical properties within the cloud have been proposed based on either a grid search of a low-dimensional cloud model (Tzabari et al., 2022) or analytic radiative transfer results for idealized cloud geometries (Davis et al., 2021). Both of these methods require cloud envelope information as input and stereo methods have shown promise at providing the required information (Dandini et al., 2022).






We showed that the use of a low accuracy forward model has been demonstrated to provide potential in accelerating retrievals. However, when angular accuracy is reduced, the retrieval problem is sufficiently non-linear that the retrievals may diverge much further than expected due to forward model error alone. This means that a low accuracy forward model is not simply a tool to get to the same vicinity of state space quicker, at least when radiance measurements are the only constraint. While its use is beneficial for the naïve initialization used here, the benefit of using a low accuracy forward model must be tested in combination with other initialization methods to determine whether it has general utility. Our results show that the choice of angular resolution of the forward model and phase function is an important potential source of diversity in retrieval accuracy. This parameter should be considered carefully in future studies of tomography, especially when comparing performance between studies which have used a wide variety of model configurations (Levis et al., 2015; Martin and Hasekamp, 2018; Levis et al., 2020; Doicu et al., 2022a, b).

In general, the ill-conditioning of the forward model in the optically thick limit indicates that radiance measurements alone are not sufficient to constrain the retrieval problem. There are many potential sources of prior information that could be incorporated into the tomographic retrieval, particularly in the optically thick limit. These may be from LES, simpler models or statistics from other measurements such as in-situ cloud probes. The important consideration when incorporating prior data is to make sure that the resulting retrieval is used to test a hypothesis for which the prior is not relevant. For this reason, input from the wider cloud and aerosol physics communities will be invaluable in guiding the development of prior constraints for physics-based inversions or training data for statistical methods in the optically thick limit. The inclusion of other measurements such as cloud radar may also be instrumental in overcoming the limitations of the technique in optically thick clouds.

One of the main motivations for developing retrievals that use 3D radiative transfer is the need to reduce the systematic biases in the retrievals that vary with the solar and viewing geometry (Marshak et al., 2006; Kato and Marshak, 2009; Di Girolamo et al., 2010; Liang and Girolamo, 2013) and the instrument resolution (Marshak et al., 2006; Zhang et al., 2012). However, our analysis identified that systematic biases can exist in the optically thick limit that are aligned with the solar direction, despite the use of 3D radiative transfer. We hypothesize that no such biases will exist for scattering regimes similar to the Thin and Medium cloud categories examined here. Testing this hypothesis and quantifying the systematic variation of the biases with solar and viewing geometry and instrument resolution is an important future step in the development and validation of the retrieval.

## 6 Summary

In this study, we have evaluated an algorithm for retrieving the 3D volumetric properties of clouds using multi-angle/multi-pixel radiances and 3D RT. The retrieval utilizes an iterative, optimization-based solution to the generalized least-squares





problem to find a best-fitting state vector parameterizing the cloud's structure. The retrieval, which was described in detail in Part 1, is publicly available in the software AT3D.


We evaluated the tomographic retrieval by applying it to synthetic measurements with a known ground truth. The synthetic measurements were generated from 10 stochastically generated, cumuliform clouds in $(1 \text{ km})^3$ domains. The extinction fields of each cloud were scaled to have maximum optical depths of 4, 17.5, 44 and 88. The 3D fields of volume extinction coefficient are retrieved at 40 m resolution, a similar resolution to the radiance measurements (35 m).


When the target clouds have maximum optical depths less than 17 the relative RMSEs and relative biases of our idealized retrievals are less than 20% and 1%, respectively. Remaining errors in these clouds are limited to the small-scale (< 250 m) spatial structure of the extinction field. The relative RMSEs in the retrieved extinction field grow with the optical depth of the cloud to reach an ensemble average of ~70% as the maximum optical thickness of the clouds reaches 88 (the value of our Very

Thick cloud category). Errors become present at larger and larger spatial scales as the optical size of the cloud increases, including a notable decrease in retrieved extinction from the illuminated to the shadowed side of the cloud and with increasing optical distance from the sensors in the optically thick clouds. This particular error pattern is attributed to the ill-posedness of the retrieval in the optically thick limit and our choice of an optically thin initialization and use of a local optimization method to solve the inverse problem.


The retrievals of clouds in the optically thick limit are highly uncertain. The addition of radiometric noise can cause large deviations in the mean extinction of individual retrievals that reach 18% for these thick clouds, although ensemble-averaged retrieval behaviour remains unchanged. The choice of angular resolution in the forward model systematically modifies the behaviour of the retrieval result for the optically thickest clouds. Using a forward model with low angular accuracy that is

better conditioned results in the reduction of ensemble-averaged relative bias in the retrievals from -36% to just -8%, which exceeds the bias improvement from forward modelling error alone (~10%). This highlights the importance of the angular accuracy in SHDOM for setting the numerical stability of the inverse problem. We suggest that using forward models with low angular accuracy is an avenue for improving the fidelity and computational efficiency of retrievals.

The tomographic retrieval's inference of optical depth outperforms an IPA-based retrieval of optical depth. The IPA retrieval, which uses nadir radiance here, has biases worse than -23%. The tomographic retrieval's biases are much smaller for the clouds with optical depths less than 44 but are comparable for the thicker clouds unless the low angular accuracy forward model is used. Relative RMSEs for the optical depth are at worst half those of the IPA.

Overall, the proposed tomographic retrieval algorithm equipped with the approximate Jacobian calculation is most promising for shallow cumulus cloud fields over near-black surfaces (e.g. ocean). These clouds have the richest available information



content within their multi-angle reflectances due to their highly heterogeneous nature. Therefore, from a fundamental perspective, these clouds are actually the easiest for passive tomographic remote sensing (Loveridge et al. 2022). Other optically thin clouds and aerosols are also likely to be effectively retrieved by this algorithm, which is an area for future work.


Deployment of the tomographic retrieval on the upcoming Cloud-CT mission (Schilling et al., 2019) has the potential to provide robust statistics of small-scale cloud properties unobtainable using in-situ measurements that are highly suitable for constraining model behaviour in climatically important shallow cumulus cloud fields. Further development of the retrieval is required to fully realize its potential of jointly retrieving aerosol and cloud properties and to improve its performance in more

difficult, optically thicker or stratiform cloud regimes. Future work will improve the realism of the retrieval to develop a full uncertainty model for the retrieval for evaluation against real observations.

**Appendix A**

The noise model used for the measurements uses the specifications from NASA's Request for Information for a Tandem Stereo instrument related to the upcoming Atmospheric Observing System mission (NASA, 2021). The Tandem Stereo instrument

would use narrowband cameras viewing in the visible at high spatial resolutions of around 50 m Ground Instantaneous Field of View per pixel. The Signal-to-Noise Ratio (SNR) requirement for this camera provides a reasonable proxy for the SNR that is achievable by an instrument that might be utilized for tomographic retrievals in practice. The SNR is defined as the ratio of the radiance, $L$, to the standard deviation of the radiance due to noise, $\sigma_L$. The SNR requirement for the Tandem Stereo camera is tabulated (Table A1) as a function of the equivalent reflectance, $R$, which is defined in terms of the band-averaged solar

irradiance, $F_0$, and the cosine of the solar zenith angle, $\mu_0$

$$R = \frac{\pi L}{\mu_0 F_0}. \tag{A1}$$

**Table A1: The Signal-to-Noise Ratio (SNR) values tabulated as a function of equivalent reflectance used to generate noise for the measurements for the tomographic retrievals. See the text for details.**

| Equivalent Reflectance | 0.01 | 0.05 | 0.1 | 0.5 | 1.0 | 1.3 |
|---|---|---|---|---|---|---|
| SNR | 87 | 201 | 285 | 639 | 904 | 1031 |

The SNR model for these measurements utilizes a cubic spline to interpolate between the values. No extrapolation at the upper end of the signal range is required. To extrapolate to reflectance values below the minimum we make the conservative assumption that the standard deviation of the radiance noise remains fixed below this reflectance level, which results in a linear decrease in the SNR till it reaches zero for no signal. This assumption is equivalent to assuming that the noise becomes dominated by dark noise below the minimum tabulated reflectance level.




The information provided is sufficient for an idealized modelling of the radiometric noise without modelling the details of the camera. We use a Poisson model to generate noise according to the SNR curve as both the photon and dark noise are well-modelled by this stochastic process and noise is typically limited by photon shot noise for most of the signal range. We neglect detailed models of other processes such as quantization, which are implicitly included in the required noise level.

The single parameter of the Poisson model, its rate, is uniquely defined as the square of the SNR. The number of counts produced in each realization of the Poisson process, $N_L$, is scaled back to a radiance by

$$L_{noisy} = \sqrt{N_L \sigma_L^2}.$$

**Appendix B**

The space carving algorithm implemented in AT3D performs a volume masking on a 3D property grid using the 2-D cloud
masks as well as the geometric information associated with each pixel. Note that the geometric information requires not just the pixel information but also how each pixel's FOV is modelled, i.e., the geometry and weights of any sub-pixel rays. See Part 1 for how this is modelled in AT3D. The algorithm then proceeds simply by tracing each ray along its line-of-sight and counting the intersections of the ray with each grid-point in the volume of the domain. An intersection with a grid-point occurs whenever a ray intersects a grid-cell for which the grid-point is one of the bounding points. The intersections of clear and
cloudy rays with each grid points are counted separately. A grid-point is designated as cloudy by a given image if the proportion of cloudy rays intersecting it for each mono-angle image is above a specified fraction (we use 0.0 for a clear conservative mask (Yang and Di Girolamo, 2008)). If the proportion of grid-point designations by each image is greater or equal to another specified fraction (typically 1.0), then the grid point is classified as cloudy. The threshold of 1.0 ensures that all grid points are designated as cloudy unless one view indicates that the grid point is clear. As such, the volume masking is clear conservative.


The accuracy of a space-carving algorithm of this sort is limited in multi-angle systems by angular resolution and spatial resolution (Lee et al., 2018). It is also limited by only using binary features (i.e., the cloud masks) to retrieve structural information and therefore can only retrieve detailed information on the cloud volume when the binary cloud masks are highly structured. Other stereoscopic approaches utilizing radiance feature matching (Seiz and Davies, 2006; Veikherman et al., 2015;
Bal et al., 2018) might perform better, especially in the case of stratiform clouds. However, in the case of cumuliform clouds, the performance of space-carving is quite good, especially at high spatial resolution, where the clear-conservative nature of the masking doesn't vastly overestimate cloud volumes.

When applied to the synthetic measurements used in the inversions, the true positive rate for grid-point classification across
all clouds is 10%, the true negative rate is 65%, the false positive rate is 25% and the false negative rate is 0%; so the true clouds take up only ~10% of the domain volume while the space carving algorithm estimates that they take up ~35%.



## Appendix C

In addition to the general setup of the inversion, we must make some specific choices about the optimization algorithm. The L-BFGS-B method is sensitive to the scaling of the problem in the initial iterations as the initial Hessian approximation in the SciPy (Virtanen et al., 2020) implementation is simply the identity matrix. The first update to the state vector is therefore simply the gradient and is therefore sensitive to the size of the initial residuals. We choose the error-covariance matrix to be a diagonal matrix with entries of $10^{-8}/m$ to compensate this effect, where $m$ is the number of measurements. With the solar flux chosen as unity (which sets the sizes of the radiances and residuals), this ensures that an initial step with a length of $\left\|\left.\frac{\partial \chi^2}{\partial \boldsymbol{a}}\right|_{\boldsymbol{a_0}}\right\|$ is not too small or too large in the sense that the Wolfe-Armijo conditions. The Wolfe-Armijo conditions on the line search use the widely-used default parameter values of $c_1 = 0.9$ and $c_2 = 10^{-4}$ (Byrd et al., 1995; Zhu et al., 1997).

The L-BFGS-B method has two key hyper-parameters. The first is the number of past iterations of state and gradient changes used to approximate the Hessian of the cost function. This is chosen to be 10 iterations. The second is the maximum number of points to search in the line search during each iteration, which is also set to 10. Three stopping conditions are also specified, the maximum number of iterations is 100, the minimum allowable relative change in the cost function before termination is set to $10^{-8}$ and the smallest maximum absolute value of the gradient before termination is also $10^{-8}$. The maximum number of iterations for the "Restarted" retrievals is still 100, in addition to the 20 to 100 iterations at low angular accuracy used to form the initialization.

## Code Availability

The software described and used in this paper is called Atmospheric Tomography with 3D Radiative Transfer (AT3D). A static archive of the software is available at Loveridge et al. (2022a). The most recent version is available from https://github.com/CloudTomography/AT3D. The original SHDOM code by Frank Evans is available from https://nit.coloradolinux.com/~evans/shdom.html.

## Competing Interests

The authors declare that they have no conflict of interest.

## Author Contributions

**JL** performed the investigation and prepared the initial draft under the supervision of **LD**. **JL**, **LD**, **YS**, **AD**, and **AL** conceptualized the study. **JL**, **AL**, **VH** and **LF** developed the software. All authors contributed to the editing of the manuscript.



**Acknowledgments**

The authors would like to thank Frank Evans for making his SHDOM code publicly available.

Jesse Loveridge was supported by NASA's FINESST program under grant agreement 80NSSC20K1633. Aviad Levis is partially supported by the Zuckerman and Viterbi postdoctoral fellowships. This research was partially carried out at the Jet Propulsion Laboratory, California Institute of Technology, under a contract with the National Aeronautics and Space Administration (80NM0018D0004). Anthony Davis was supported by the ROSES NRA Program Element TASNPP17-0165.
Support from the MISR project through the Jet Propulsion Laboratory of the California Institute of Technology under Contract 1474871 for Larry Di Girolamo is gratefully acknowledged. Linda Forster was funded by the European Union's Framework Programme for Research and Innovation Horizon 2020 (2014–20) under the Marie Skłodowska-Curie Grant Agreement 754388 (LMUResearchFellows) and from LMUexcellent, funded by the Federal Ministry of Education and Research (BMBF) and the Free State of Bavaria under the Excellence Strategy of the German Federal Government and the
Länder. Yoav Schechner is the Mark and Diane Seiden Chair in Science at the Technion. He is a Landau Fellow - supported by the Taub Foundation. His work was conducted in the Ollendorff Minerva Center. Minerva is funded through the BMBF. This project has received funding from the European Union's Horizon 2020 research and innovation programme under grant agreement No 810370-ERC-CloudCT. The authors are grateful to the US-Israel Binational Science Foundation (BSF grant 2016325) for facilitating our international collaboration.

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
