# Peer review of "Retrieving 3D distributions of atmospheric particles using Atmospheric Tomography with 3D Radiative Transfer – Part 2: local optimization"

_Atmospheric Measurement Techniques, 2023_

## Referee Comment (RC1)

**Review of Loveridge et al. paper "Retrieving 3D distributions of atmospheric particles using Atmospheric Tomography with 3D Radiative Transfer – Part 2: local optimization" submitted to AMT**

The paper evaluates the tomographic retrieval described in detail in part 1. The software package Atmospheric Tomography with 3D Radiative Transfer (AT3D) is publicly available. In the paper AT3D applies to synthetic measurements (10 stochastically generated cumuliform clouds) with a known truth. This is a fundamental paper that deserves to get published in AMT. I enjoyed reading it, and I am sure it will be well cited. The only concern I have is that this manuscript is too long and requires a lot of time and affords to get through. I would recommend dividing it in two parts: methods and results. It will help to get more readers. Anyway, it is up to the authors to make the decision.

**Some Minor Suggestions**

Pg. 2. Give a reference to Evans (1998) when mentioned SHDOM the first time.

Pg. 4-5. Comparing Eqs. (1) and (3), I don't see the derivative d$R$/da. How to compute approximate Jacobian? Why does it accuracy degrade with more isotropic phase function?

Pg. 6. Please provide an example of the smallest and the largest singular values for a typical ill-conditioned (optically thick clouds) problem.

Pg. 10. Stochastically generated clouds used to be more popular, e.g., Cahalan et al., 1989; Lovejoy, 1992.

Pg. 11. Did you use other that -5/3 exponents? Provide a reference, e.g., *Lovejoy, S., D. Schertzer, P. Silas, Y. Tessier, D. Lavallée, 1993: The unified scaling model of the atmospheric dynamics and systematic analysis of scale invariance in cloud radiances. Annales Geophysicae, 11, 119-127.*

Pg. 12. I would recommend replacing Fig. 2b with a more informative one, at least with color scale.

Pg. 14. Please define $\|\cdot\|_1$ and $\|\cdot\|_2$ in Eqs. (6) and (7).

Pg. 14. Please define again what is *Noisy-GT* here or use *Noisy-GroundTruth* instead.

---

## Author Response (AR1)

**Reviewer #1**

We would like to thank the reviewer for taking the time to carefully review the paper.

> 1) *It took me a few reads back and forth to understand what is meant when stated that the approximate (low angular accuracy) forward model is "better conditioned" and why this would be. If I understand correctly from pages 5 and 7 it is because the approximate Jacobian is more appropriate for the approximate forward model. I suggest that the authors repeat this explanation on page 9, section 4.1, the summary and the abstract. Using the term "better conditioned' will not be understood by every reader.*

We have replaced the term "better conditioned" with the term "less ill-conditioned" throughout the manuscript. We have clarified that "less ill-conditioned" simply means "smaller condition number" in both the Abstract and at the end of Page 7. We have also explicitly stated the distinction between "appropriateness" or accuracy of the approximate Jacobian and its conditioning to ensure these concepts don't get confused on Page 8:

"The reduction in condition number of the forward model with decreasing angular resolution, the increase in the accuracy of the approximate Jacobian with decreasing angular resolution stem from the same cause but have different effects on a local optimization procedure. Ill-conditioning is the sensitivity of the inversion to errors. The approximation to the Jacobian is a source of error."

> 2) *I am wondering whether this approach can benefit from the approximate tomography approach presented by Alexandrov et al. (2021) to improve the first guess and therefore the convergence.*

As we discuss in our paper, initialization methods may indeed be helpful to improve performance. In principle, the Alexandrov et al. (2021) method can provide such an initialization. However, there are some issues to consider. We have added a small note providing more detail about potential initialization strategies (including Alexandrov et al.) in the Discussion section at the end of Page 31 (Line 738).

"Other methods to retrieve the internal extinction field are also possible such as using 1D radiative transfer or heuristics based on 1D radiative transfer and linear tomography (Alexandrov et al., 2021). Both such methods neglect the asymmetry in the radiance field between forward and backward scattering geometries, which will lead to overestimation of the extinction field on the illuminated side of the cloud. An initialization with such an artefact would likely encourage the formation of the local minimum observed here in optically thick clouds with a gradient in extinction from the illuminated to shadowed side. Further study is needed to identify the most effective initialization methods and their range of applicability in terms of solar zenith angle and cloud optical depth."

**Reviewer #2**

We would like to thank the reviewer for taking the time to carefully review the manuscript.

> *The only concern I have is that this manuscript is too long and requires a lot of time and affords to get through. I would recommend dividing it in two parts: methods and results. It will help to get more readers. Anyway, it is up to the authors to make the decision.*

We acknowledge that the manuscript is detailed and on the long side. As this paper is already Part 2 of a work, with no clear way in dividing this into Part 2 and 3 without affecting the natural flow of the work, we have opted not to divide the paper further. There are many papers in AMT that are longer (e.g., Kotthaus et al. 2023), so we are not setting any precedent here.

Kotthaus, S., Bravo-Aranda, J. A., Collaud Coen, M., Guerrero-Rascado, J. L., Costa, M. J., Cimini, D., O'Connor, E. J., Hervo, M., Alados-Arboledas, L., Jiménez-Portaz, M., Mona, L., Ruffieux, D., Illingworth, A., and Haeffelin, M.: Atmospheric boundary layer height from ground-based remote sensing: a review of capabilities and limitations, Atmos. Meas. Tech., 16, 433–479, https://doi.org/10.5194/amt-16-433-2023, 2023

> *Pg. 2. Give a reference to Evans (1998) when mentioned SHDOM the first time.*

Done.

> *Pg. 4-5. Comparing Eqs. (1) and (3), I don't see the derivative dR/da.*

We have added dR/da to Eq. 3, though we don't give details on how it is computed as it will depend on the choice of regularization and such terms are generally constructed so that they have a tractable linearization.

> *How to compute approximate Jacobian? Why does it accuracy degrade with more isotropic phase function?*

Details of how to compute the approximate Jacobian are given in Part 1, as well as a detailed explanation of the dependence of the accuracy on the phase function. We have added a sentence summarizing the reasoning at Line 125 and references to Part 1 for these specific statements:

"This is due to the larger relative contributions of higher-order scattering to the gradients, which are the contributions which are least accurately modelled in our approximate Jacobian calculation (Loveridge et al., 2023)."

> *Pg. 6. Please provide an example of the smallest and the largest singular values for a typical ill-conditioned (optically thick clouds) problem.*

We have included examples for our configuration (0.06 and 2e-7). However, these are not general numbers, as we have noted in the text. The magnitude of the Jacobian for derivatives of radiance with respect to optical properties also varies with grid resolution. Finer grids make all of

the singular values smaller. The absolute magnitude of the source for the radiative transfer also affects this. However, the condition number is largely invariant to these factors and describes the inverse problem well.

> *Pg. 10. Stochastically generated clouds used to be more popular, e.g., Cahalan et al., 1989; Lovejoy, 1992.*
>
> *Pg. 11. Did you use other that -5/3 exponents?*
> *Provide a reference, e.g.,*
>
> *Lovejoy, S., D. Schertzer, P. Silas, Y. Tessier, D. Lavallée, 1993: The unified scaling model of the atmospheric*
> *dynamics and systematic analysis of scale invariance in cloud radiances. Annales Geophysicae, 11, 119-127.*

We have clarified that we only used an exponent of *-5/3* in our work.
Thank you for the suggested references. We have added this reference and others regarding the use of stochastic models and the evidence for the scaling behavior in observations.

> *Pg. 12. I would recommend replacing Fig. 2b with a more informative one, at least with color scale.*

We have modified this qualitative visualization by adding an image calculated using 3D radiative transfer to give a sense of perspective as well as adding the accompanying optical path and maximum volume extinction coefficient along the line of sight of the image for quantitative interpretation.

> *Pg. 14. Please define $|| . ||_1$ and $|| . ||_2$ in Eqs. (6) and (7).*

We have included definitions.

> *Pg. 14. Please define again what is Noisy-GT here or use Noisy-GroundTruth instead.*

We have used Noisy-GroundTruth throughout the manuscript for clarity.